# List-Decodable Sparse Mean Estimation

**Shiwei Zeng**
Department of Computer Science
Stevens Institute of Technology
szeng4@stevens.edu

**Jie Shen**
Department of Computer Science
Stevens Institute of Technology
jie.shen@stevens.edu

## Abstract

Robust mean estimation is one of the most important problems in statistics: given a set of samples in $\mathbb{R}^d$ where an $\alpha$ fraction are drawn from some distribution $D$ and the rest are adversarially corrupted, we aim to estimate the mean of $D$. A surge of recent research interest has been focusing on the list-decodable setting where $\alpha \in (0, \frac{1}{2}]$, and the goal is to output a finite number of estimates among which at least one approximates the target mean. In this paper, we consider that the underlying distribution $D$ is Gaussian with $k$-sparse mean. Our main contribution is the first polynomial-time algorithm that enjoys sample complexity $O\big(\mathrm{poly}(k, \log d)\big)$, i.e. poly-logarithmic in the dimension. One of our core algorithmic ingredients is using low-degree *sparse polynomials* to filter outliers, which may find more applications.

## 1 Introduction

Mean estimation is arguably a fundamental inference task in statistics and machine learning. Given a set of samples $\{x_1, \ldots, x_n\} \subset \mathbb{R}^d$ where an $\alpha$ fraction are drawn from some well-behaved (e.g. Gaussian) distribution $D$ and the rest are adversarially corrupted, the goal is to estimate the mean of $D$. In the noiseless case where $\alpha = 1$, the problem can be easily solved in view of the concentration of measure phenomenon [LT91]. However, this is rarely the case as modern data sets are often contaminated by random noise or even by adversarial corruptions. Thus, a great deal of recent efforts are focused on efficiently and robustly estimating the target mean in the presence of outliers.

Generally speaking, there is a phase transition between $\alpha > 1/2$ and $0 < \alpha \le 1/2$, and solving either problem in a computationally efficient manner is highly nontrivial. The problem that most of the samples are uncorrupted, i.e. $\alpha > 1/2$, has a very long history dating back to the 1960s [Tuk60, Hub64], yet only until recently have computationally efficient algorithms been established [DKK$^+$16, LRV16]. The other yet more challenging regime concerns that an overwhelming fraction of the samples are corrupted, i.e. $\alpha \le 1/2$, which even renders estimation impossible. This motivates a line of research on *list-decodable* mean estimation [CSV17], where in place of outputting one single estimate, the algorithm is allowed to generate a finite list of candidates and is considered to be successful if there exists at least one candidate in the list that is sufficiently close to the target mean.

In this work, we investigate the problem of list-decodable mean estimation, for which there have been a plethora of elegant results established in recent years. From a high level, most of them concern error guarantees and running time. For example, [CSV17] proposed the first tractable algorithm based on semidefinite programming, which runs in polynomial time and achieves optimal error rate for variance-bounded distributions. [DKS18b] developed a multi-filtering scheme and showed that the error rate can be improved by using high degree polynomials if the underlying distribution is Gaussian. The more recent works [CMY20, DKK$^+$21a] further addressed the computational efficiency of this task and achieved almost linear running time in certain regimes.

Although all of these algorithms exhibit near-optimal guarantees on either error rate or computational complexity, it turns out that less is explored to improve another yet important metric: the sample

complexity. In particular, the sample complexity of all these algorithms is $O(\text{poly}(d))$, hence they quickly break down for data-demanding applications such as healthcare where the number of available samples is typically orders of magnitude less than the dimension $d$ [Wai19]. Therefore, a pressing question that needs to be addressed in such a high-dimensional regime is the following:

> Does there exist a provably robust algorithm for list-decodable mean estimation that runs in polynomial time and enjoys a sample complexity bound of $O(\text{polylog}(d))$?

In this paper, we answer the question in the affirmative by showing that when the target mean is $k$-sparse, i.e. it has at most $k$ non-zero elements, it is *attribute-efficiently* list-decodable.

**Theorem 1** (Main result). *Given parameter $\alpha \in (0, \frac{1}{2}]$, failure probability $\tau \in (0,1)$, a natural number $\ell \geq 1$, and a set $T$ of $\Omega\left(\frac{\ell^4 \cdot k^{8\ell}}{\alpha^7} \cdot \log^{6\ell}\left(\frac{\ell d}{\alpha \tau}\right)\right)$ samples in $\mathbb{R}^d$, of which at least a $(2\alpha)$-fraction are independent draws from the Gaussian distribution $N(\mu, \mathbb{I}_d)$ where $\|\mu\|_0 \leq k$, there exists an algorithm that runs in time $\text{poly}\left(|T|, d^\ell, \frac{1}{\alpha}\right)$, uses polynomials of degree at most $2\ell$, and returns a list of $O(1/\alpha)$ number of $k$-sparse vectors such that with probability $1 - \tau$, the list contains at least one $\hat{\mu} \in \mathbb{R}^d$ with $\|\hat{\mu} - \mu\|_2 = \tilde{O}(\alpha^{-\frac{1}{2\ell}} \cdot \sqrt{\ell}(\ell + \log \frac{1}{\alpha}))$, where $\tilde{O}(\cdot)$ hides poly-logarithmic factors.*

**Remark 2.** The key message of the theorem is that when the true mean is $k$-sparse, it is possible to efficiently approximate it with $O(\text{polylog}(d))$ samples. This is in stark contrast to existing list-decodable results [CSV17, DKS18b, CMY20, DKK20a, DKK+21a] where the sample complexity is $O(\text{poly}(d))$. The only attribute-efficient robust mean estimators are [BDLS17, DKK+19, CDK+21], but their results hold only for the mild corruption regime where $\alpha > 1/2$.

**Remark 3.** Our algorithm and analysis hold for any degree $\ell \geq 1$. When $\ell = 1$, the sample complexity reads as $\tilde{O}(\alpha^{-7} k^8 \log^6 d)$ and the algorithm achieves error $\tilde{O}(\alpha^{-\frac{1}{2}})$. As opposed to an $O(1 - \alpha)$ error rate obtained for $\alpha > 1/2$, the (non-vanishing) error rate $\tilde{O}(\alpha^{-\frac{1}{2}})$ is typically what one can expect for list-decodable mean estimation under bounded second order moment condition, in light of the lower bounds in [DKS18b]. When leveraging degree-$2\ell$ polynomials into algorithmic design, we obtain the improved $\tilde{O}\left(\alpha^{-\frac{1}{2\ell}}\sqrt{\ell}(\ell + \log \frac{1}{\alpha})\right)$ error guarantee. Specially, when taking $\ell = \Theta\left(\log \frac{1}{\alpha}\right)$, our algorithm achieves error rate of $\tilde{O}\left(\log^{\frac{3}{2}}\left(\frac{1}{\alpha}\right)\right)$ in quasi-polynomial time. This is very close to the minimax error rate of $\Theta(\log^{\frac{1}{2}}\left(\frac{1}{\alpha}\right))$ established in [DKS18b].

**Remark 4.** If we further increase the sample size with an $\ell^\ell$ multiplicative factor with $\ell = \Theta(\log \frac{1}{\alpha})$, our algorithm will achieve an $\tilde{O}\left(\log^{\frac{1}{2}}\left(\frac{1}{\alpha}\right)\right)$ error guarantee, which matches the minimax lower bound. The proof follows the same pipeline and we leave it to interested readers.

## 1.1 Overview of Our Techniques

Our main algorithm is inspired by the multifiltering framework of [DKS18b], where the primary idea is to construct a sequence of polynomials to test the concentration of the samples to Gaussian so that the algorithm either certifies that the sample set behaves like Gaussian, or sanitizes it by removing a sufficient amount of outliers. Our key technical contribution lies into a new design of *sparse polynomials*, and new filtering rules tailored to the sparse polynomials.

**Sparse polynomials and sparsity-induced filters.** To ensure that our algorithm is attribute-efficient, we will only control the maximum eigenvalue of the sample covariance matrix on sparse directions. Since such computation is NP-hard in general, we first consider a sufficient condition which tests the maximum Frobenuis norm under a cardinality constraint, similar to the idea of [DKK+19]. If such Frobenuis norm is small, it implies a small restricted eigenvalue and hence the sample mean is returned. Otherwise, we construct sparse polynomials in the sense that they can be represented by a set of $O(\ell^2 k^{4\ell})$ basis polynomials and $O(\ell k^{2\ell})$ coordinates of the samples (see Definition 7), and measure the concentration of these sparse polynomials to the Gaussian. Now as the underlying polynomials are sparse, we also design new sparsity-induced filters to certify the sample set, as otherwise a large amount of clean samples will be removed. See Algorithm 3 and Algorithm 4.

**Clustering by $L_\infty$-norm.** Technically, the success of our attribute-efficient multifiltering approach hinges on a condition that all the samples lie within a small $L_\infty$-norm ball. It is not hard to see that all the Gaussian samples satisfy such condition, and we show that there is a simple scheme which can simultaneously prune and cluster the given samples into $O(1/\alpha)$ groups, such that the

retained samples are close under the $L_\infty$-norm and at least one group contains most of the Gaussian samples. We note that the use of the $L_\infty$-norm as our metric ensures attribute efficiency of this step. An immediate implication of this clustering step is that the polynomials of Gaussian samples will be close enough, which facilitates the analysis of the performance of our filters. See Section 2.3.

## 1.2 Related Works

Breaking the barrier of the typical $O(\text{poly}(d))$ sample complexity bound is one of the central problems across many fields of science and engineering. Motivated by real-world applications, a property termed sparsity is often assumed for this end, meaning that only $k$ out of the $d$ number of attributes contribute to the underlying inference problem. In this way, an improved bound of $O(\text{poly}(k, \log d))$ can be obtained in many inference paradigms such as linear regression [CDS98, Tib96, CT05, Don06, SL17a, SL17b, SL18, WSL18], learning of threshold functions [Lit87, BHL95, STT12, PV13, ABHZ16, ZSA20, She20, SZ21], principal component analysis [Ma13, DKK+19], and mean estimation [BDLS17, DKK+19, CDK+21]. Unfortunately, the success of all these attribute-efficient algorithms hinges on the presumption that the majority of the data are uncorrupted.

**Learning with mild corruption** ($\alpha > 1/2$). Learning in the presence of noise has been extensively studied in a broad context. In supervised learning where a sample consists of an instance (i.e. feature vector) and a label, lots of research efforts were dedicated to robust algorithms under label noise [AL87, Slo88, MN06]. Recent years have witnessed significant progress towards optimal algorithms in the presence of label noise, see for example, [KKMS05, ABL17, DKTZ20, ZSA20, DKK+20b, ZS22] and the references therein. The regime that both instances and labels are corrupted turns out to be significantly more challenging. The problem of learning halfspaces under such setting was put forward in the 1980s [Val85, KL88], yet only until recently have efficient algorithms been established with near-optimal noise tolerance [ABL17, DKS18a, She21, SZ21]. In addition, [BJK15, KKM18, LSLC20] studied robust linear regression and [BDLS17] presented a set of interesting results under various statistical models. More in line with this work is the problem of robust mean estimation, see the breakthrough works of [DKK+16, LRV16] and many follow-up works [DKK+17, BDLS17, DKS17, SCV18, KSS18, DKK+19, HLZ20, CDK+21].

**Learning with overwhelming corruption** ($\alpha \le 1/2$). The agnostic label noise of [Hau92, KSS92] seems the earliest model that allows the adversary to arbitrarily corrupt any fraction of the data (say 70%), though it can only corrupt labels. Following [CSV17], a considerate number of of recent works have studied the scenario that both instances and labels are grossly corrupted, and the goal is to output a finite list of candidate parameters among which at least one is a good approximation to the target. This includes list-decodable learning of mixture models [DKS18b, DKK+21b], regression [KKK19, RY20a], and subspace recovery [RY20b, BK21]. Interestingly, there are some works studying the problem under crowdsourcing models, where the samples are collected from crowd workers and most of them behave adversarially [SVC16, ABHM17, MV18, ZS21].

It is worth noting that [DKK+22] concurrently and independently developed a polynomial-time algorithm to solve the same problem, with an interesting difference-of-pairs metric to filter outliers.

## 1.3 Roadmap

We collect useful notations, definitions, and some preliminary results in Section 2. Our main algorithms are described in Section 3 along with performance guarantees. We conclude the work in Section 4, and defer all proof details to the appendix.

## 2 Preliminaries

**Vector, matrix, and tensor.** For a $d$-dimensional vector $v = (v_1, \ldots, v_d)$, denote by $\|v\|_2$ its $L_2$-norm, $\|v\|_1$ its $L_1$-norm, $\|v\|_0$ its $L_0$-"norm" that counts the number of non-zeros, and $\|v\|_\infty$ its infinity norm. The hard thresholding operator $\text{trim}_k : \mathbb{R}^d \to \mathbb{R}^d$ keeps the $k$ largest elements (in magnitude) of a vector and sets the remaining to zero. Let $[d] := \{1, 2, \ldots, d\}$ for some natural number $d > 0$. For an index set $\Omega \subseteq [d]$, $v_\Omega$ is the vector of $v$ restricted on $\Omega$. We say a vector is $k$-sparse if it has at most $k$ non-zero elements, and likewise for matrices and tensors. For a matrix $M$ of size $d_1 \times d_2$, denote by $\|M\|_F$ its Frobenius norm and by $\|M\|_*$ its nuclear norm. For $U \subseteq [d_1] \times [d_2]$, denote by $M_U$ the submatrix of $M$ with entries restricted to $U$.

We also use tensors in our algorithms to ease expressions. Note that vectors and matrices can be seen as order-1 and order-2 tensors respectively. We say that an order-$l$ tensor $A$ is symmetric if $A_{i_1,\ldots,i_l} = A_{\pi(i_1,\ldots,i_l)}$ for all permutations $\pi$. Given two tensors $A$ and $B$, denote by $A \otimes B$ the outer product (or tensor product) of $A$ and $B$. We will slightly abuse $\|A\|_2$ to denote the $L_2$-norm of a tensor $A$ by seeing it as a long vector.

**Probability.** We reserve the capital letter $G$ for a random draw from $N(\mu, \mathbb{I}_d)$, i.e. $G \sim N(\mu, \mathbb{I}_d)$, where $\mu \in \mathbb{R}^d$ is the target mean that we aim to estimate which is assumed to be $k$-sparse. Suppose that $T$ is a finite sample set. We use $\mu_T$ to denote the sample mean of $T$, i.e. $\mu_T = \frac{1}{|T|} \sum_{x \in T} x$, and use $p(T)$ to denote the random variable $p(x)$ where $x$ is drawn uniformly from $T$.

**Constants.** The capital letter $C$ and its subscript variants such as $C_1, C_2$ are used to denote positive absolute constants. However, their values may change from appearance to appearance.

## 2.1 Polynomials

Let $x = (x_1, \ldots, x_d)$ be a $d$-dimensional vector in $\mathbb{R}^d$, and let $\boldsymbol{a} = (\boldsymbol{a}_1, \ldots, \boldsymbol{a}_d) \in \mathbb{N}^d$ be a $d$-dimensional multi-index. A *monomial* of $x$ is a product of powers of the coordinates of $x$ with natural exponents, written as $x^{\boldsymbol{a}} := \prod_{j=1}^d x_j^{\boldsymbol{a}_j}$. A *polynomial* of $x$, $p(x)$, is a finite sum of its monomials multiplied by real coefficients; that is, $p(x) = \sum_{\boldsymbol{a} \in \mathcal{A}} c_{\boldsymbol{a}} x^{\boldsymbol{a}}$ where $\mathcal{A} \subset \mathbb{N}^d$ is a finite set of multi-indices and the $c_{\boldsymbol{a}}$'s are real coefficients. Note that the degree of $p(x)$ is given by $\max_{\boldsymbol{a} \in \mathcal{A}} \|\boldsymbol{a}\|_1$. We denote by $\mathbb{P}(\mathbb{R}^d, l)$ the class of polynomials on $\mathbb{R}^d$ with degree at most $l$. We will often use the probabilist's Hermite polynomials that form a complete orthogonal basis with respect to $N(0, \mathbb{I}_d)$.

**Definition 5** (Hermite polynomials). Let $x \in \mathbb{R}$ be a variate. For any natural number $l \in \mathbb{N}$, the degree-$l$ Hermite polynomial is defined as $\mathrm{He}_l(x) = (-1)^l e^{\frac{x^2}{2}} \frac{d^l}{dx^l} e^{-\frac{x^2}{2}}$. For $\boldsymbol{a} \in \mathbb{N}^d$ and $x \in \mathbb{R}^d$, the $d$-variate Hermite polynomial is given by $\mathrm{He}_{\boldsymbol{a}}(x) := \prod_{i=1}^d \mathrm{He}_{\boldsymbol{a}_i}(x_i)$, which is of degree $\|\boldsymbol{a}\|_1$.

**Harmonic and homogeneous polynomials.** A polynomial $h(x) \in \mathbb{P}(\mathbb{R}^d, l)$ is called *harmonic* if it can be written as a linear combination of degree-$l$ Hermite polynomials. A polynomial $\mathrm{Hom}(x) \in \mathbb{P}(\mathbb{R}^d, l)$ is called *homogeneous* if all of its monomials have degree exactly $l$.

**Fact 6.** If a polynomial is degree-$l$ harmonic or homogeneous, then there is a one-to-one mapping between it and an order-$l$ symmetric tensor.

To see this, we may define an operation "$\circ$" such that $\mathrm{He}_l(x_i) \circ \mathrm{He}_l(x_j) = \mathrm{He}_l(x_i) \cdot \mathrm{He}_l(x_j)$ if $i \neq j$ and equals $\mathrm{He}_{2l}(x_i)$ otherwise. Then any degree-$l$ Hermite polynomial can be written as $\mathrm{He}_1(x_{i_1}) \circ \mathrm{He}_1(x_{i_2}) \cdots \circ \mathrm{He}_1(x_{i_l})$ where all the indices $i_t \in [d]$. We will consider that one such sequence $(i_1, \ldots, i_l)$ exactly corresponds to one degree-$l$ Hermite polynomial on $\mathbb{R}^d$, and there are $d^l$ number of such sequences that form all degree-$l$ Hermite polynomials. In this sense, any harmonic polynomial $h(x)$ can be written as $h(x) = \sum_{i_1,\ldots,i_l} A_{i_1,\ldots,i_l} \cdot \mathrm{He}_1(x_{i_1}) \circ \mathrm{He}_1(x_{i_2}) \cdots \circ \mathrm{He}_1(x_{i_l})$, where $A_{i_1,\ldots,i_l}$'s are the coefficients which form an order-$l$ tensor. If we choose $A$ as symmetric, it is easy to see that $A$ fully represents $h(x)$. Then, we can convert "$\circ$" back to the regular product by counting the number of times a particular index $j$ appearing in $(i_1, \ldots, i_l)$. If we denote this number as $c_j(i_1, \ldots, i_l)$, we have

$$h(x) = \frac{1}{\sqrt{l!}} \sum_{i_1,\ldots,i_l} A_{i_1,\ldots,i_l} \prod_j \mathrm{He}_{c_j(i_1,\ldots,i_l)}(x_j) =: h_A(x), \text{ with } \sum_{j=1}^d c_j(i_1,\ldots,i_l) = l, \quad (2.1)$$

where the factor $1/\sqrt{l!}$ is only used to normalize the magnitude of $A$ to ease our analysis.

Likewise, any homogeneous polynomial takes the form

$$\mathrm{Hom}_A(x) = \sum_{i_1,\ldots,i_l} A_{i_1,\ldots,i_l} \prod_j x_j^{c_j(i_1,\ldots,i_l)}.$$

**Sparse polynomials.** In order to define sparse polynomials, we will first specify a set of basis polynomials $\{b_1, \ldots, b_{d^l}\} \subset \mathbb{P}(\mathbb{R}^d, l)$. In this paper, we will either choose such set as all degree-$l$ monomials or all degree-$l$ Hermite polynomials.

**Definition 7** $((\kappa, \psi)$-sparse polynomials). We say that a polynomial $p \in \mathbb{P}(\mathbb{R}^d, l)$ is $(\kappa, \psi)$-sparse if it can be represented by at most $\kappa$ number of basis polynomials and $\psi$ coordinates of the input vector. We denote by $\mathbb{P}(\mathbb{R}^d, l, \kappa, \psi)$ the class of $(\kappa, \psi)$-sparse polynomials.

Note that when $\kappa$ and $l$ are fixed, $p(x)$ will depend on at most $\kappa \cdot l$ coordinates. Thus, the introduction of the parameter $\psi$ makes sense only when $\psi \leq \kappa \cdot l$. In our algorithm, we will always have $l \leq 2\ell$, $\kappa = 4\ell^2 k^{4\ell}$, and $\psi = 2\ell k^{2\ell}$ for some natural number $\ell \geq 1$.

## 2.2 Representative Set and Good Set

To ease our analysis, we will need a deterministic condition on the set of uncorrupted samples.

**Definition 8** (Representative set). Given $\alpha \in (0, \frac{1}{2}]$ and $\tau \in (0, 1)$, we say that a sample set $S_G \subset \mathbb{R}^d$ is representative with respect to $\mathbb{P} := \mathbb{P}(\mathbb{R}^d, 2\ell, 4\ell^2 k^{4\ell}, 2\ell k^{2\ell})$ if the following holds:

$$\sup_{p \in \mathbb{P}} |\Pr[p(G) \geq 0] - \Pr[p(S_G) \geq 0]| \leq \epsilon_0, \text{ where } \epsilon_0 := \frac{\alpha^3}{100 k^{2\ell} \cdot \log^{2\ell}(\frac{\ell d}{\alpha \tau})}.$$

We show that a sufficiently large set drawn independently from $N(\mu, \mathbb{I}_d)$ is representative. The proof follows from the classic VC theory, and is deferred to Appendix A.1.

**Proposition 9** (Sample complexity). *Given $\alpha \in (0, \frac{1}{2}]$ and $\tau \in (0, 1)$, let $S_G$ be a set consisting of $|S_G| = C \cdot \frac{(l \cdot \kappa + \psi) \log d}{\epsilon^2} \log \frac{(l \cdot \kappa + \psi) \log d}{\epsilon \tau}$ independent samples from $N(\mu, \mathbb{I}_d)$ where $C > 0$ is a sufficiently large absolute constant. Then, with probability $1 - \tau$,*

$$\sup_{p \in \mathbb{P}(\mathbb{R}^d, l, \kappa, \psi)} |\Pr[p(G) \geq 0] - \Pr[p(S_G) \geq 0]| \leq \epsilon.$$

*In particular, when $l = 2\ell$, $\kappa = 4\ell^2 k^{4\ell}$, $\psi = 2\ell k^{2\ell}$, and $\epsilon = \frac{\alpha^3}{100 k^{2\ell} \cdot \log^{2\ell}(\frac{\ell d}{\alpha \tau})}$ for some natural number $\ell \geq 1$, it suffices to pick $|S_G| = C' \cdot \frac{\ell^4 \cdot k^{8\ell}}{\alpha^6} \cdot \log^{6\ell}(\frac{\ell d}{\alpha \tau})$ for some sufficiently large constant $C'$ so that $S_G$ is a representative set.*

Our algorithm will progressively remove samples from $T$, and a key property that ensures the success of the algorithm is that most corrupted samples are eliminated while almost all uncorrupted samples are retained. Alternatively, we hope that $T$ contains a representative set that contributes to a nontrivial fraction. For technical reasons, we also require that all samples in $T$ lie in a small $L_\infty$-ball.

**Definition 10** ($\alpha$-good set). A multiset $T \subset \mathbb{R}^d$ is $\alpha$-good if the following holds:

1. There exists a set $S_G$ which is representative and satisfies $|S_G \cap T| \geq \max\{(1 - \alpha/6)|S_G|, \alpha|T|\}$.

2. $\max_{x,y \in T} \|x - y\|_\infty \leq C \cdot \sqrt{\log(d|S_G|/\tau)}$ for some constant $C > 0$.

It is not hard to verify that the initial sample set $T$ satisfies the first condition, and will also fulfill the second one with a simple data pre-processing, as stated in the next section.

## 2.3 Clustering for the Initial List

Since the corrupted samples may behave adversarially, we will perform a preliminary step of clustering which splits $T$ into an initial list of subsets, among which at least one is $\alpha$-good in the sense of Definition 10. We first show that all Gaussian samples have bounded $L_\infty$-norm with high probability, which simply follows from the Gaussian tail bound.

**Lemma 11.** *Given $\tau \in (0, 1)$, with probability $1 - \tau$, we have $\max_{x \in S_G} \|x - \mu\|_\infty \leq \sqrt{2 \log \frac{d|S_G|}{\tau}}$, where $S_G$ is a set of samples drawn independently from $N(\mu, \mathbb{I}_d)$.*

The above observation implies that for any $x, y \in S_G$, their distance under the $L_\infty$-norm metric is at most $2\sqrt{2 \log(d|S_G|/\tau)} \leq O(\sqrt{\ell \cdot \log \frac{\ell d}{\alpha \tau}})$ as far as the size of $S_G$ has the same order with the one in Proposition 9. To guarantee the existence of such $S_G$, it suffices to draw a corrupted sample set $T$ that is $1/\alpha$ times larger than $|S_G|$. The lemma below further shows that this is sufficient to guarantee the existence of an $\alpha$-good subset of $T$.

**Algorithm 1** CLUSTER$(T, \alpha, \tau, \ell)$

---

**Require:** A multiset of samples $T \subset \mathbb{R}^d$, parameter $\alpha \in (0, 1/2]$, failure probability $\tau \in (0, 1)$, degree of polynomials $\ell \geq 1$.

1: A set of centers $\mathcal{C} \leftarrow \emptyset$, radius $\gamma \leftarrow C_0 \cdot \sqrt{\ell \cdot \log \frac{\ell d}{\alpha \tau}}$ for some constant $C_0 > 0$.
2: For each $x \in T$, proceed as follows: **if** there are at least $\alpha \cdot |T|$ samples $y$ in $T$ that satisfy $\|x - y\|_\infty \leq 2\gamma$, and no sample $x' \in \mathcal{C}$ satisfies $\|x - x'\|_\infty \leq 6\gamma$ **then** $\mathcal{C} \leftarrow \mathcal{C} \cup \{x\}$.
3: For each $x_i \in \mathcal{C}$, let $T_i = \{y \in T : \|x_i - y\|_\infty \leq 6\gamma\}$.
4: **return** $\{T_1, \ldots, T_{|\mathcal{C}|}\}$.

---

**Lemma 12** (CLUSTER). *Given $\alpha \in (0, \frac{1}{2}]$ and $\tau \in (0, 1)$, let $T$ be the sample set given to the learner. If $|T| = C \cdot \frac{\ell^7 \cdot k^{14\ell}}{\alpha^7} \cdot \log^{8\ell}(\frac{\ell d}{\alpha \tau})$ and a $(2\alpha)$-fraction are independent samples from $N(\mu, \mathbb{I}_d)$, Algorithm 1 returns a list of at most $1/\alpha$ many subsets of $T$, such that with probability at least $1 - \tau$, at least one of them is an $\alpha$-good set.*

As will be clear in our analysis, the motivation of bounding the $L_\infty$-distance is to make sure that the function value of any $p(x) = h_A(x - \mu_T) \in \mathbb{P}(\mathbb{R}^d, l, \kappa, \psi)$ is bounded for samples in the $\alpha$-good subset $T_i$. This is because when there exist a significant fraction of good samples in $T_i$, we want to efficiently distinguish the corrupted and uncorrupted ones. A value-bounded polynomial function will facilitate our analysis on the function variance.

**Lemma 13.** *Suppose that $T$ is $\alpha$-good and a polynomial $p \in \mathbb{P}(\mathbb{R}^d, l, 4\ell^2 k^{4\ell}, 2\ell k^{2\ell})$ satisfies the following: there exists a symmetric order-l tensor $A$ such that $\|A\|_2 \leq 1$ and $p(x) = h_A(x - \mu_T)$.*

*Then, it holds that $\max_{x,y \in T} |p(x) - p(y)| \leq 2k^\ell \cdot \gamma^l$, where $\gamma = C_0 \cdot \sqrt{\ell \cdot \log(\frac{\ell d}{\alpha \tau})}$.*

## 3 Main Algorithms and Performance Guarantees

We start with a review of the multifiltering framework that has been broadly used in prior works [DKS18b, DKK20a, DKK+21b], followed by a highlight of our new techniques.

The multifiltering framework, i.e. Algorithm 2, includes three major steps. The first step is to invoke CLUSTER (Algorithm 1) to generate an initial list $\mathcal{L}$ which guarantees the existence of an $\alpha$-good subset of $T$ (see Lemma 12). We then imagine that there is a tree with root being the original contaminated sample set $T$ and each child node of the root represents a member in $\mathcal{L}$. The algorithm iterates through these child nodes and performs one of the following: (1) creating a leaf node which is an estimate of the target mean; (2) creating one or two child nodes where are subsets of the parent node; (3) certifying that the set cannot be $\alpha$-good and delete branch. In the end, if all leaves of the tree cannot be further split or deleted, the mean of the subsets on leaf nodes will be collected as a list $M$. It is worth noting that the goal of algorithmic design is to guarantee that there always exists a branch that includes only $\alpha$-good subsets. In other words, at any level of the algorithm, at least one of the subsets of $T$ is $\alpha$-good, which ensures the existence of a good estimation in the returned list $M$. The final step is a black-box algorithm that reduces the size of $M$ from $O(\text{poly}(1/\alpha))$ to $O(1/\alpha)$, which is due to [DKS18b]. Our technical contributions lie into an attribute-efficient implementation of the first and second steps. In this section, we elaborate on the second step, i.e. the ATTRIBUTE-EFFICIENT-MULTIFILTER algorithm.

### 3.1 Overview of Attribute-Efficient Multifiltering

The ATTRIBUTE-EFFICIENT-MULTIFILTER algorithm is presented in Algorithm 3. The starting point of the algorithm is a well-known fact that if the adversary were to significantly deteriorate our estimate on $\mu$, the spectral norm of a certain sample covariance matrix $\tilde{\Sigma}$ would become large [DKK+16, LRV16]. In order to achieve attribute-efficient sample complexity $O(\text{poly}(k, \log d))$, it is however vital to control the spectral norm only on $k^\ell$-sparse directions for some pre-specified polynomial degree $\ell \geq 1$, which can further be certified by a small Frobenius norm restricted on the largest $k^{2\ell}$ entries. If the restricted Frobenius norm is sufficiently small, it implies that the sample covariance matrix behaves as a Gaussian one, and the algorithm returns the empirical mean truncated to be $k$-sparse (see Step 4). Otherwise, the algorithm will invoke either BASICMF (i.e. Algorithm 4)

---

**Algorithm 2** Main Algorithm: Attribute-Efficient List-Decodable Mean Estimation

---

**Require:** A multiset of samples $T \subset \mathbb{R}^d$, parameter $\alpha \in (0, 1/2]$, failure probability $\tau \in (0, 1)$, degree of polynomials $\ell \geq 1$.

1: $\{T_1, \ldots, T_m\} \leftarrow \text{CLUSTER}(T, \alpha, \tau, \ell)$, $\mathcal{L} \leftarrow \{(T_1, \alpha/2), \ldots, (T_m, \alpha/2)\}$, $M \leftarrow \emptyset$.
2: **while** $\mathcal{L} \neq \emptyset$ **do**
3:     $(T', \alpha') \leftarrow$ an element in $\mathcal{L}$, $\mathcal{L} \leftarrow \mathcal{L} \backslash \{(T', \alpha')\}$.
4:     $\text{ANS} \leftarrow \text{ATTRIBUTE-EFFICIENT-MULTIFILTER}(T', \alpha', \tau/|T|, \ell)$.
        (i)    **if** ANS is a vector **then** add it into $M$.
        (ii)   **if** ANS is a list of $(T_i, \alpha_i)$ **then** append those with $\alpha_i \leq 1$ to $\mathcal{L}$.
        (iii)  **if** ANS = NO **then** go to the next iteration.
5: **end while**
6: **return** $\text{LISTREDUCTION}(T, \alpha, \ell, M)$.

---

---

**Algorithm 3** ATTRIBUTE-EFFICIENT-MULTIFILTER$(T, \alpha, \tau, \ell)$

---

**Require:** A multiset of samples $T \subset \mathbb{R}^d$, parameter $\alpha \in (0, 1/2]$, failure probability $\tau \in (0, 1)$, degree of polynomials $\ell \geq 1$.

1: $\tilde{\Sigma} \leftarrow \mathbb{E}[P_{d,\ell}(T - \mu_T) \cdot P_{d,\ell}(T - \mu_T)^\top]$, and $P_{d,\ell}(x)$ is the column vector of all degree-$\ell$ Hermite polynomials of $x$.
2: $\{(i_t, j_t)\}_{t \geq 1}^{\frac{1}{2}(k^{2\ell}+k^\ell)} \leftarrow$ index set of the $k^\ell$ diagonal entries and $\frac{1}{2}(k^{2\ell} - k^\ell)$ entries above the main diagonal of $\tilde{\Sigma}$ with largest magnitude. $U \leftarrow \{(i_t, j_t)\}_{t \geq 1} \cup \{(j_t, i_t)\}_{t \geq 1}$, $U' \leftarrow I \times I$, with $I = \{i_t\}_{t \geq 1} \cup \{j_t\}_{t \geq 1}$.
3: $\lambda^*_{\text{sparse}} \leftarrow \left[ C_1 \cdot (\ell + C_1 \log \frac{1}{\alpha}) \cdot \log^2(2 + \log \frac{1}{\alpha}) \right]^{2\ell}$ for large enough constant $C_1 > 0$.
4: **if** $\left\| (\tilde{\Sigma})_U \right\|_F \leq \lambda^*_{\text{sparse}}$ **then return** $\hat{\mu} \leftarrow \text{trim}_k(\mu_T)$.
5: $(\lambda^*, v^*) \leftarrow$ the largest eigenvalue and eigenvector of $(\tilde{\Sigma})_{U'}$.
6: **if** $\lambda^* \geq \lambda^*_{\text{sparse}}$ **then**
7:     **if** $\ell = 1$ **then** $\text{ANS} \leftarrow \text{BASICMF}(T, \alpha, \tau, p_1)$ **else** $\text{ANS} \leftarrow \text{HARMONICMF}(T, \alpha, \tau, p_1)$ where $p_1(x) := v^* \cdot P_{d,\ell}(x - \mu_T)$.
8: **else**
9:     $p_2(x) \leftarrow \frac{1}{\|A'\|_F} \cdot \left( P_{d,\ell}(x - \mu_T)^\top \cdot A' \cdot P_{d,\ell}(x - \mu_T) \right)$ with $A' := (\tilde{\Sigma})_{U'}$.
10:    $\text{ANS} \leftarrow \text{HARMONICMF}(T, \alpha, \tau, p_2)$.
11: **end if**
12: **return** ANS.

---

or HARMONICMF (i.e. Algortihm 5) to examine the concentration of a polynomial of the empirical data to that of Gaussian. Both algorithms will either assert that the current sample set does not contain a sufficiently large amount of Gaussian samples, or will prune many corrupted samples to increase the fraction of Gaussian ones. A more detailed description of the two algorithms can be found in Section 3.2.1 and Section 3.2.2 respectively. What is subtle in Algorithm 3 is that we will check the maximum eigenvalue $\lambda^*$ of the empirical covariance matrix $\tilde{\Sigma}$ restricted on a carefully chosen subset $U'$, which corresponds to the maximum eigenvalue on a certain $(2k^{2\ell})$-sparse direction. If $\lambda^*$ is too large, this indicates an *easy* problem since it must be the case that the adversary corrupted the samples in an aggressively way. Therefore, it suffices to prune outliers using a degree-$\ell$ polynomial $p_1$ which is simply the projection of $P_{d,\ell}(x - \mu_T)$ onto the span of the maximum eigenvector; see Step 7 in Algorithm 3. On the other hand, if $\lambda^*$ is on a moderate scale, it indicates that the adversary corrupted the samples in a very delicate way so that it passes the tests of both Frobenius norm and spectral norm. Now the main idea is to check the concentration of higher degree polynomials induced by the sample set; we show that it suffices to construct a degree-$2\ell$ harmonic polynomial; see Step 10.

While sparse mean estimation has been studied in [DKK+19] and the idea of using restricted Frobenius norm and filtering was also developed, we note that their analysis only holds in the mild corruption regime where $\alpha > 1/2$. To establish the main results, we will leverage the tools from [DKS18b], with a specific treatment on the fact that $\mu$ is $k$-sparse, to ensure an attribute-efficient sample complexity bound. As we will show later, a key idea to this end is to utilize a sequence of carefully chosen sparse polynomials in the sense of Definition 7 along with sparsity-induced filters.

The performance guarantee of ATTRIBUTE-EFFICIENT-MULTIFILTER is as follows.

**Theorem 14** (Algorithm 3). *Consider Algorithm 3 and denote by* ANS *its output. With probability* $1 - \tau$, *the following holds.* ANS *cannot be* TBD. *If* ANS *is a $k$-sparse vector and if $T$ is $\alpha$-good,* *then* $\|\mu - \hat{\mu}\|_2 \leq \tilde{O}\big(\alpha^{-\frac{1}{2\ell}} \sqrt{\ell}(\ell + \log \frac{1}{\alpha})\big)$. *If* ANS = NO, *then $T$ is not $\alpha$-good. If* ANS = $\{(T_i, \alpha_i)\}_{i=1}^m$ *for some $m \leq 2$, then $T_i \subset T$ for all $i \in [m]$ and $\sum_{i=1}^m \frac{1}{\alpha_i^2} \leq \frac{1}{\alpha^2}$; if additionally $T$ is* $\alpha$-good, *then at least one $T_i$ is $\alpha_i$-good. Finally, the algorithm runs in time $O\big(\text{poly}(|T|, d^\ell)\big)$.*

### 3.2 Analysis of ATTRIBUTE-EFFICIENT-MULTIFILTER

We first show that if the restricted Frobenius norm of the sample covariance matrix is small, then the sample mean is a good estimate of the target mean.

**Lemma 15.** *Consider Algorithm 3. If the algorithm returns a vector $\hat{\mu}$ at Step 4 and if $T$ is $\alpha$-good,* *we have that* $\|\hat{\mu} - \mu\|_2 \leq O\big(\alpha^{-\frac{1}{2\ell}} \sqrt{\ell} \cdot (\ell + \log \frac{1}{\alpha}) \cdot \log^2(2 + \log \frac{1}{\alpha})\big)$.

Next, we give performance guarantees on the remaining steps of Algorithm 3, where we consider the case that the algorithm does not return at Step 4. Namely, the algorithm will either reach at Step 7 or Step 10, and will return the ANS obtained thereof. These two steps will invoke BASICMF or HARMONICMF on different sparse polynomials. Observe that both algorithms may return 1) "NO", which certifies that the current input set $T$ is not $\alpha$-good; 2) a list of subsets $\{(T_i, \alpha_i)\}_{i=1}^m$ for some $m \leq 2$, on which Algorithm 3 will be called in a recursive manner; or 3) TBD, which indicates that the algorithm is uncertain on $T$ being $\alpha$-good. In the following, we prove that the way that we invoke BASICMF and HARMONICMF ensures that they will never return TBD when being called within Algorithm 3. We then give performance guarantees on these two filtering algorithms when they return "NO" or $\{(T_i, \alpha_i)\}_{i=1}^m$, thus establishing Theorem 14.

Let us consider that the algorithm reaches Step 7, i.e. the largest eigenvalue on one sparse direction is larger than the threshold $\lambda^*_{\text{sparse}}$. It is easy to see that when $\ell = 1$, ANS cannot be TBD since the only way that BASICMF returns TBD is when $\text{Var}[p(T)]$ is not too large, but this would violate the condition that $\lambda^* > \lambda^*_{\text{sparse}}$ in view of our setting on $\lambda^*_{\text{sparse}}$. Similarly, we show that under the large $\lambda^*$ regime, HARMONICMF will not return TBD either. Thus, we have the following lemma.

**Lemma 16.** *Consider Algorithm 3. If it reaches Step 7, then* ANS $\neq$ TBD.

Now it remains to consider the case that the algorithm reaches Step 10, which is more subtle since the evidence from the magnitude of the largest restricted eigenvalue is not so strong to prune outliers. Note that this could happen even when $T$ contains many outliers, since $\lambda^*$ is not the maximum eigenvalue on all sparse directions but on a submatrix indexed by $U'$. Fortunately, if $\lambda^*$ is not large, we show that the algorithm can still make progress by calling HARMONICMF on degree-$2\ell$ sparse polynomials. This is because higher-degree polynomials are more sensitive to outliers than low-degree polynomials, as far as we can certify the concentration of high-degree polynomials on clean samples. As a result, we will have the following guarantee.

**Lemma 17.** *Consider Algorithm 3. If it reaches Step 10, then* ANS $\neq$ TBD.

#### 3.2.1 Basic Multifilter for Sparse Polynomials

The BASICMF algorithm (Algorithm 4) is a key ingredient in the multifiltering framework. It takes as input a sparse polynomial $p$ and uses it to certify whether $T$ is $\alpha$-good and sufficiently concentrated. The central idea is to measure how $p(T)$ distributed and compare it to that of the distribution of $p(G)$. We require the input $p$ has certifiable variance on $G$, i.e. $\text{Var}[p(G)] \leq 1$, as otherwise, it could filter away a large number of the good samples. We note that the bounded variance condition is always satisfied for degree-1 Hermite polynomials under proper normalization, while for high-degree polynomials, one cannot invoke BASICMF directly (see Section 3.2.2 for a remedy).

The way that BASICMF certifies the input sample set $T$ not being $\alpha$-good is quite simple: if not all samples lie in a small $L_\infty$-ball, it returns "NO" at Step 2, in that this contradicts Lemma 13. Otherwise, the algorithm will attempt to search for a finer interval $[a, b]$ such that it includes most of the samples. If such interval exists, then either the adversary corrupted the samples such that the sample variance is as small as that of Gaussian while the sample mean may deviate far from the target, in which case BASICMF returns TBD at Step 5; or the sample variance is large, in which case

---

**Algorithm 4** BASICMF$(T, \alpha, \tau, p)$

---

**Require:** A multiset of samples $T \subset \mathbb{R}^d$, parameter $\alpha \in (0, 1/2]$, failure probability $\tau \in (0, 1)$, a polynomial $p \in \mathbb{P}(\mathbb{R}^d, l, 4\ell^2 k^{4\ell}, 2\ell k^{2\ell})$ such that $l \leq 2\ell$, $\mathrm{Var}[p(G)] \leq 1$, and $p(x) = h_A(x - \mu_T)$.

1: $R \leftarrow (C_1 \cdot \log \frac{1}{\alpha})^{l/2}$, $\gamma \leftarrow C_0 \cdot \sqrt{\ell \cdot \log \frac{\ell d}{\alpha \tau}}$.
2: **if** $\max_{x,y \in T} |p(x) - p(y)| > 2k^\ell \cdot \gamma^l$ **then return** "NO".
3: **if** there is an interval $[a, b]$ of length $C_1 \cdot R \cdot \log(2 + \log \frac{1}{\alpha})$ that contains at least $(1 - \frac{\alpha}{2})$-fraction of samples in $\{p(x) : x \in T\}$ **then**
4:     **if** $\mathrm{Var}[p(T)] \leq C_1 \cdot (\ell + C_1 \log \frac{1}{\alpha})^l \cdot \log^2(2 + \log \frac{1}{\alpha})$ **then**
5:         **return** "TBD".
6:     **else**
7:         Find a threshold $t > 2R$ such that

$$\Pr_{x \sim T}\left[\min\{|p(x) - a|, |p(x) - b|\} \geq t\right] > \frac{32}{\alpha} \exp(-(t - 2R)^{2/l}) + \frac{2\alpha^2}{k^{2\ell} \log^l(\frac{\ell d}{\alpha \tau})}.$$

8:         $T' \leftarrow \{x \in T : \min\{|p(x) - a|, |p(x) - b|\} \leq t\}$, $\alpha' \leftarrow \alpha \cdot \left(\frac{(1 - \alpha/8)|T|}{|T'|} + \frac{\alpha}{8}\right)$.
9:         **return** $\{(T', \alpha')\}$.
10:     **end if**
11: **else**
12:     Find $t \in \mathbb{R}$, $R' > 0$ such that the sets $T_1 := \{x \in T : p(x) > t - R'\}$ and $T_2 := \{x \in T : p(x) < t + R'\}$ satisfy

$$|T_1|^2 + |T_2|^2 \leq |T|^2 (1 - \alpha/100)^2 \text{ and } |T| - \max(|T_1|, |T_2|) \geq \alpha |T| / 4.$$

13:     $\alpha_i \leftarrow \alpha \cdot (1 - \alpha^2/100) \cdot |T| / |T_i|$, for $i = 1, 2$.
14:     **return** $\{(T_1, \alpha_1), (T_2, \alpha_2)\}$.
15: **end if**

---

it is possible to construct a sparsity-induced filter to prune outliers (see Steps 7 and 8). We note that in Step 7, the first term on the right-hand side is derived from Chernoff bound for degree-$l$ Gaussian polynomials and the second term is due to concentration of empirical samples to Gaussian (see Definition 8), both of which are scaled by a factor $8/\alpha$ so that the number of the samples removed from $T$ is $8/\alpha$ times more than that of the good samples in the representative set $S_G \subset T$, which means most of the removed samples are outliers. We show by contradiction the existence of the threshold $t$ (see Lemma 26). In fact, had such threshold $t$ not existed, the set $T$ must be sufficiently concentrated such that the algorithm would have returned at Step 5. This essentially relies on our result of the initial clustering of Algorithm 1, which guarantees that each subset $T$ is bounded in a small $L_\infty$-ball and the function value of $p$ on the $\alpha$-good $T$ does not change drastically (Lemma 13). We then show that equipped with such threshold $t$, $T'$ is a subset of $T$ and it is $\alpha'$-good if $T$ is $\alpha$-good (Lemma 28).

When there is no such short interval $[a, b]$, the algorithm splits $T$ into two overlapping subsets $\{T_1, T_2\}$ such that $T_1 \cap T_2$ is large enough to contain most of the samples in $S_G$. This guarantees that most of the samples in $S_G$ (if $T$ is $\alpha$-good) are always contained in one subset and thus there always exists an $\alpha$-good subset of $T$. We show that an appropriate threshold $t$ can also be found at Step 12 (Lemma 30), and at least one $T_i$ is $\alpha_i$-good if $T$ is $\alpha$-good.

As a result, we have the following guarantees for Algorithm 4; see Appendix B for the full proof.

**Theorem 18** (BASICMF)**.** *Consider Algorithm 4. Denote by* ANS *its return. Suppose that $T$ being $\alpha$-good implies $\mathrm{Var}[p(G)] \leq 1$. Then with probability $1 - \tau$, the following holds.* ANS *is either "NO", "TBD", or a list of $\{(T_i, \alpha_i)\}_{i=1}^m$ with $m \leq 2$. 1) If* ANS = NO*, then $T$ is not $\alpha$-good. 2) If* ANS = TBD*, then $\mathrm{Var}[p(T)] \leq O\big((\ell + \log \frac{1}{\alpha})^l \cdot \log^2(2 + \log \frac{1}{\alpha})\big)$; and if additionally $T$ is $\alpha$-good, then $|\mathbb{E}[p(G)] - \mathbb{E}[p(T)]| \leq O\big((\ell + \log \frac{1}{\alpha})^{\frac{l}{2}} \cdot \log(2 + \log \frac{1}{\alpha})\big)$. 3) If* ANS $= \{(T_i, \alpha_i)\}_{i=1}^m$*, then $T_i \subset T$ and $\sum_i \frac{1}{\alpha_i^2} \leq \frac{1}{\alpha^2}$ for all $i \in [m]$; if additionally $T$ is $\alpha$-good, then at least one $T_i$ is $\alpha_i$-good.*

---
**Algorithm 5** HARMONICMF$(T, \alpha, \tau, p)$

---

**Require:** A multiset of samples $T \subset \mathbb{R}^d$, parameter $\alpha \in (0, 1/2]$, failure probability $\tau \in (0, 1)$, a polynomial $p \in \mathbb{P}(\mathbb{R}^d, l, 2\ell k^{2\ell}, 2\ell k^{2\ell})$ such that $p(x) = h_A(x - \mu_T)$ and $\|A\|_2 = 1$.

1: **for** $l' = 0, 1, \ldots, l$ **do**
2:     Let $B^{(l')}$ be an order-$2l'$ tensor with

$$B^{(l')}_{i_1, \ldots, i_{l'}, j_1, \ldots, j_{l'}} = \sum_{k_{l'+1}, \ldots, k_l} A_{i_1 \ldots, i_{l'}, k_{l'+1}, \ldots, k_l} A_{j_1 \ldots, j_{l'}, k_{l'+1}, \ldots, k_l}.$$

3:     Consider $B^{(l')}$ as a $d^{l'} \otimes d^{l'}$ symmetric matrix by grouping each of the $i_1, \ldots, i_{l'}$ and $j_1, \ldots, j_{l'}$ coordinates together. Apply eigenvalue decomposition on $B^{(l')}$ to obtain $B^{(l')} = \sum_i \lambda_i V_i \otimes V_i$.
4:     $\text{ANS}_i \leftarrow$ MULTILINEARMF$(T, V_i, l', \alpha, \tau/(ld^l))$ for every $V_i$. If $\text{ANS}_i = \text{NO}$ or a list of $\{(T_j, \alpha_j)\}$ for some $i$, then **return** $\text{ANS}_i$. If $\text{ANS}_i = \text{TBD}$, **continue**.
5: **end for**
6: $\text{ANS} \leftarrow$ BASICMF$(T, \alpha, \tau, \frac{1}{\beta} h_A(x - \mu_T))$ with $\beta := \left( C_1 \cdot (1 + \log \frac{1}{\alpha}) \cdot \log^2(2 + \log \frac{1}{\alpha}) \right)^{\frac{l}{2}}$.
    If $\text{ANS} = \text{NO}$ or a list of $(T_j, \alpha_j)$, **return** $\text{ANS}$. If $\text{ANS} = \text{TBD}$, still **return** "NO".

---

### 3.2.2 Harmonic Multifilter with Hermite Polynomials

Recall that applying BASICMF (Algorithm 4) on a polynomial $p$ requires $\text{Var}[p(G)] \leq 1$. It is nontrivial to verify this condition for a high-degree polynomial $p$, as the variance of high-degree Gaussian polynomials depends on the distribution mean, i.e. $\mu - \mu_T$ in this case, which is unfortunately unknown. As a remedy, notice that for any harmonic polynomial $h_A(x)$, $\mathbb{E}_{x \sim N(\mu', \mathbb{I})}[h_A(x)^2]$ equals the summation of homogeneous polynomials of $\mu'$, which can also be seen as the expectation of multilinear polynomials over independent variables $X_{(i)} \sim N(\mu', \mathbb{I}_d)$. Thus, we only need to verify the expectation of these corresponding multilinear polynomials, whose variance on $G$ does not hinge on $\mu'$. The harmonic multifilter is presented in Algorithm 5, where the subroutine MULTILINEARMF can be found in Appendix D. We first present the guarantee when Algorithm 5 returns all TBD at Step 4 and reaches Step 6, where we can certify a bounded variance for $h_A(x - \mu_T)$ on $G$.

**Lemma 19** (Variance of $p$). *Consider Algorithm 5. If it reaches Step 6 and $T$ is $\alpha$-good, then we have $\mathbb{E}[h_A(G - \mu_T)^2] \leq \beta^2$.*

Based on Lemma 19, we have that $\text{Var}[h_A(G - \mu_T)/\beta] \leq 1$, for which we can invoke BASICMF on $h_A(x - \mu_T)/\beta$ and Theorem 18 can be applied immediately. We are ready to elaborate the proof ideas for Lemma 16 and 17. First, observe that BASICMF returns "TBD" at Step 6 if and only if $\text{Var}[h_A(T - \mu_T)/\beta] \leq C_1 \cdot \left( \ell + C_1 \log \frac{1}{\alpha} \right)^l \cdot \log^2(2 + \log \frac{1}{\alpha})$. Now return to Algorithm 3. When $h_A(x - \mu_T) = p_1(x)$, this could not happen because $\text{Var}[p_1(T)] = \text{Var}[v^* \cdot P_{d,\ell}(T - \mu_T)] \geq \lambda^* \geq \lambda^*_{\text{sparse}} = \left[ C_1 \cdot (\ell + C_1 \log \frac{1}{\alpha}) \cdot \log^2(2 + \log \frac{1}{\alpha}) \right]^{2\ell} \geq \beta^2 \cdot C_1 \cdot \left( \ell + C_1 \log \frac{1}{\alpha} \right)^{\ell} \cdot \log^2(2 + \log \frac{1}{\alpha})$. A contradiction that implies Lemma 16. When $h_A(x - \mu_T) = p_2(x)$, the case is more delicate. Here, we instead show that $T$ must not be $\alpha$-good and HARMONICMF will return "NO" correctly. This is because if $T$ is $\alpha$-good, Proposition 18 implies that $\mathbb{E}[p_2(G)]$ is close to $\mathbb{E}[p_2(T)]$, and together with Lemma 19 we can show that $\mathbb{E}[p_2(T)]$ is small. However, by construction $\left\| (\hat{\Sigma})_U \right\| = \mathbb{E}[p_2(T)] \geq \lambda^*_{\text{sparse}}$, a contradiction that gives Lemma 17. The detailed proof can be found in Appendix B.3.

## 4 Conclusion and Future Work

In this paper, we developed an attribute-efficient mean estimation algorithm which achieves sample complexity poly-logarithmic in the dimension with low-degree sparse polynomials under the list-decodable setting. A natural question is whether the current techniques could be utilized to attribute-efficiently solve the other list-decodable problems, such as learning of halfspaces and linear regression.

## Acknowledgments and Disclosure of Funding

We thank the anonymous reviewers and meta-reviewer for valuable discussions. This work is supported by NSF-IIS-1948133 and the startup funding from Stevens Institute of Technology.

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
