# A    Omitted Proofs from Section 2

## A.1    Proof of Proposition 9

*Proof.* Fix a subset $\Omega \subset [d]$ with size $\psi$, and then fix a set of $\kappa$ monomials on $\Omega$ with degree at most $l$, denoted by $\mathcal{M}(\Omega, l)$. Let $\mathbb{P}(\mathbb{R}^d, \mathcal{M}(\Omega, l), \Omega)$ be the induced class of polynomials. Note that $\mathbb{P}(\mathbb{R}^d, l, \kappa, \psi) = \cup_\Omega \cup_{\mathcal{M}(\Omega,l)} \mathbb{P}(\mathbb{R}^d, \mathcal{M}(\Omega, l), \Omega)$.

It is easy to see that for any $p \in \mathbb{P}(\mathbb{R}^d, \mathcal{M}(\Omega, l), \Omega)$, it can be represented by a linear combinations of the $\kappa$ monomials. Thus, the VC dimension of this class equals $\kappa + 1$. Then, we note that there are $\sum_{j=0}^{\psi} \binom{d}{j}$ choices of $\Omega$, and for any given $\Omega$, there are $\sum_{j=0}^{\kappa} \binom{2d^l}{j}$ choices of $\mathcal{M}(\Omega, l)$. Therefore, the total number of the subclass $\mathbb{P}(\mathbb{R}^d, \mathcal{M}(\Omega, l), \Omega)$ is at most

$$\sum_{j=0}^{\psi} \binom{d}{j} \cdot \sum_{j=0}^{\kappa} \binom{2d^l}{j} \leq \left(\frac{ed}{\psi}\right)^\psi \cdot \left(\frac{2ed^l}{\kappa}\right)^\kappa. \tag{A.1}$$

The concept class union argument states that for $\mathcal{H} = \cup_{i=1}^m \mathcal{H}_i$, the VC dimension of $\mathcal{H}$ is upper bounded by $O(\max\{V, \log m + V \log \frac{\log m}{V}\})$, where $V$ is an upper bound on the VC dimension of all $\mathcal{H}_i$. In our case, we have $V = \kappa + 1$ and $m \leq \left(\frac{ed}{\psi}\right)^\psi \cdot \left(\frac{2ed^l}{\kappa}\right)^\kappa$. By calculation, we can show that the VC dimension of $\mathbb{P}(\mathbb{R}^d, l, \kappa, \psi)$ is upper bounded by

$$\psi \log \frac{ed}{\psi} + \kappa \log \frac{2ed^l}{\kappa} + \kappa + 1 \leq (l\kappa + \psi) \log d =: d'. \tag{A.2}$$

Recall that the VC theory states that for any $\epsilon, \tau \in (0, 1)$, as long as $|S_G| \geq C \left(\frac{d'}{\epsilon^2} \log \frac{d'}{\epsilon} + \frac{1}{\epsilon^2} \log \frac{1}{\tau}\right)$ for some absolute constant $C > 0$, the following holds with probability $1 - \tau$:

$$\sup_{p \in \mathbb{P}(\mathbb{R}^d, l, \kappa, \psi)} \left| \Pr[p(G) \geq 0] - \Pr_{x \sim S_G}[p(x) \geq 0] \right| \leq \epsilon. \tag{A.3}$$

With the expression of $d'$ in (A.2), it is not hard to see that we can set $|S_G| = C \cdot \frac{(l \cdot \kappa + \psi) \log d}{\epsilon^2} \log \frac{(l \cdot \kappa + \psi) \log d}{\epsilon \tau}$ for some absolute constant $C > 0$ to ensure that the above holds.

When $l = 2\ell$, $\kappa = 4\ell^2 k^{4\ell}$, $\psi = 2\ell k^{2\ell}$, and $\epsilon = \frac{\alpha^3}{100k^{2\ell} \cdot \log^{2\ell}(\frac{\ell d}{\alpha \tau})}$ for some natural number $\ell \geq 1$, by algebraic calculation, it suffices to pick $|S_G| = C' \cdot \frac{\ell^4 \cdot k^{8\ell}}{\alpha^6} \cdot \log^{6\ell}(\frac{\ell d}{\alpha \tau})$ for some sufficiently large constant $C'$. This completes the proof. $\qquad\square$

## A.2    Proof of Lemma 11

*Proof.* By the standard tail bound of Gaussian distribution, for any $x$ drawn from $N(\mu, \mathbb{I}_d)$, it holds that for any given index $i \in [d]$, $\Pr[|x_i - \mu_i| \geq t] \leq 2 \exp(-t^2/2)$. By taking union bound over both index $i$ and sample $x \in S_G$, we have $\Pr[\max_{x \in S_G} \max_{i \in [d]} |x_i - \mu_i| \geq t] \leq 2d |S_G| \exp(-t^2/2)$. Choosing $t = \sqrt{2 \log(d |S_G| / \tau)}$ completes the proof. $\qquad\square$

## A.3    Proof of Lemma 12

*Proof.* Let $S_G$ be the subset of $T$ containing the samples drawn i.i.d. from $N(\mu, \mathbb{I}_d)$. Since $|S_G| = 2\alpha \cdot |T|$, we know that $S_G$ is a representative set with probability at least $1 - \tau$ in light of Prop. 9.

Consider Algorithm 1. If for all $x, y \in T$, we have

$$\|x - y\|_\infty \leq 6\gamma, \tag{A.4}$$

then the algorithm returns only one cluster and the lemma follows immediately.

If that is not the case, we first note that, with probability at least $1 - \tau$ all of the samples in $S_G$ satisfy Eq. (A.4) due to Lemma 11. Let us condition on this event occurs from now on.

Algorithm 1 constructs a set of disjoint $L_\infty$-balls of radius $2\gamma$, of which each is centered at one sample in $T$ and contains at least an $\alpha$-fraction of samples in $T$. Therefore, the number of such balls is at most $m = \lfloor 1/\alpha \rfloor$. Denote the set by $\{\mathbb{B}_1, \ldots, \mathbb{B}_m\}$. Let $\mathbb{B}'_i$ be the ball that has the same center as $\mathbb{B}_i$ but with $\ell_\infty$-radius of $6\gamma$. In the following, we show that there exists $i \in [m]$, such that $T_i = T \cap \mathbb{B}'_i$ is $\alpha$-good.

Consider a sample $x \in S_G$, for which we know that $\|x - \mu\|_\infty \le \gamma$. Then, for the $L_\infty$-ball $\mathbb{B}_x := \{y \in \mathbb{R}^d : \|y - x\|_\infty \le 2\gamma\}$, all of the samples in $S_G$ will be contained in $\mathbb{B}_x$. In addition, there must exist one $\mathbb{B}_i$ that intersects $\mathbb{B}_x$, as otherwise $\mathbb{B}_x$ will be in the set $\{\mathbb{B}_1, \ldots, \mathbb{B}_m\}$. That is, $\exists z \in T, z \in \mathbb{B}_x \cap \mathbb{B}_i$. By construction, $\mathbb{B}_x$ must be containted in $\mathbb{B}'_i$. Therefore, all samples of $S_G$ must be included in $T_i$ and $T_i$ is $\alpha$-good. $\square$

### A.4 Proof of Lemma 13

*Proof.* Recall that after running Algorithm 1, every subset $T_i$ is contained in an $L_\infty$-ball of radius $6\gamma$. By Jensen's inequality and the convexity of the $L_\infty$-norm, we have for all $x \in T$, $\|x - \mu_T\|_\infty \le 6\gamma$.

Recall that we assumed $p(x) = h_A(x - \mu_T)$. Thus $\mathbb{E}_{x \sim N(\mu_T, \mathbb{I})}[p(x)] = 0$ due to the definition of harmonic polynomials. Thus, $\text{Var}_{x \sim N(\mu_T, \mathbb{I})}[p(x)] = \|A\|_2^2$. Denote $z = x - \mu_T$. Then,

$$|p(x)| = \left| \sum_{j \in [k^{2\ell}]} c_{\boldsymbol{a}^{(j)}} \frac{\text{He}_{\boldsymbol{a}^{(j)}}(z)}{\sqrt{\|\boldsymbol{a}^{(j)}\|_1!}} \right| \le \sqrt{\left( \sum_{j \in [k^{2\ell}]} c_{\boldsymbol{a}^{(j)}}^2 \right) \left( \sum_{j \in [k^{2\ell}]} \frac{\text{He}_{\boldsymbol{a}^{(j)}}(z)^2}{\|\boldsymbol{a}^{(j)}\|_1!} \right)}. \tag{A.5}$$

where $\boldsymbol{a}^{(j)}$ is a $d$-dimensional multi-index for the $j$-th monomial, and $c_{\boldsymbol{a}^{(j)}}$ denotes its coefficient. Observe that in the first step, $p(x)$ is written as a linear combination of $k^{2\ell}$ Hermite polynomials, since we are considering $p \in \mathbb{P}(\mathbb{R}^d, l, k^{2\ell}, 2\ell k^{2\ell})$. Note also that $\sum_{j \in [k^{2\ell}]} c_{\boldsymbol{a}^{(j)}}^2 = \|A\|_2^2 \le 1$.

To bound the second factor on the right-hand side of (A.5), we use Mehler's formula, which shows that for any $u$ with $|u| < 1$ and any natural number $a$,

$$\sum_{a=0}^{\infty} \frac{\text{He}_a^2(z_i) u^a}{a!} = \frac{1}{\sqrt{1 - u^2}} e^{\frac{u}{1+u} z_i^2},$$

Since each $\text{He}_{\boldsymbol{a}^{(j)}}(z)$ has degree at most $l$, it can be decomposed as a product of at most $l$ univariate Hermite polynomials. Thus, we take such product and sum over $j \in [k^{2\ell}]$ to obtain

$$\sum_{j \in [k^{2\ell}]} \frac{\prod_{\boldsymbol{a}_i^{(j)} \neq 0} \left( \text{He}_{\boldsymbol{a}_i^{(j)}}(z_i)^2 \cdot u^{\boldsymbol{a}_i^{(j)}} \right)}{\|\boldsymbol{a}^{(j)}\|_1!} \le k^{2\ell} \cdot (1 - u^2)^{-\frac{l}{2}} \cdot e^{\frac{u}{1+u} \|\text{trim}_l(z)\|_2^2}.$$

To simplify the above expression, observe that $\prod_{\boldsymbol{a}_i^{(j)} \neq 0} u^{\boldsymbol{a}_i^{(j)}} = u^{\|\boldsymbol{a}^{(j)}\|_1} \ge u^l$. In addition, $\|\text{trim}_l(z)\|_2^2 \le l \cdot \|z\|_\infty^2 \le 36l\gamma^2$. Lastly, by algebra, $(1 - u^2)^{-\frac{l}{2}} \le e^{\frac{u^2 l}{2}}$. Putting all pieces together gives

$$\sum_{j \in [k^{2\ell}]} \frac{\text{He}_{\boldsymbol{a}^{(j)}}(z)^2}{\|\boldsymbol{a}^{(j)}\|_1!} \le u^{-l} \cdot k^{2\ell} \cdot e^{\frac{u^2 l}{2}} \cdot e^{\frac{36l\gamma^2 u}{1+u}} = k^{2\ell} \cdot u^{-l} \cdot e^{\frac{u^2 l}{2} + \frac{36l\gamma^2 u}{1+u}}.$$

We set $u = \frac{1}{\gamma}$; this is possible as $\gamma > 1$. Then the exponent $\frac{u^2 l}{2} + \frac{36l\gamma^2 u}{1+u} = \frac{l}{2\gamma^2} + \frac{36l}{1 + 1/\gamma^2} \le 37l$. Without loss of generality, we may assume that $\gamma > e^{37}$; in fact, we can always ensure this by setting $\gamma = (C_0 + e^{37}) \cdot \sqrt{\ell \cdot \log \frac{\ell d}{\alpha \tau}}$ where $C_0$ is the constant given in Algorithm 1. Thus, it follows that

$$\sum_{j \in [k^{2\ell}]} \frac{\text{He}_{\boldsymbol{a}^{(j)}}(z)^2}{\|\boldsymbol{a}^{(j)}\|_1!} \le k^{2\ell} \cdot \gamma^l \cdot e^{37l} \le k^{2\ell} \cdot \gamma^{2l}.$$

Plugging it into (A.5) completes the proof. $\square$

# B  Analysis of ATTRIBUTE-EFFICIENT-MULTIFILTER

We collect a few useful facts about Hermite polynomials.

Recall that for an order-$l$ tensor $A \in \mathbb{R}^{d^l}$, $\|A\|_2$ denotes its $L_2$ norm by seeing it as a long vector, and for a polynomial $p : \mathbb{R}^d \to \mathbb{R}$, $\|p\|_2 := \mathbb{E}_{x \sim N(0, \mathbb{I}_d)}[p^2(x)]^{1/2}$.

The following can be easily seen from the definition of harmonic polynomials.

**Fact 20.** For all order-$l$ symmetric tensors $A$ and its corresponding harmonic polynomial $h_A$, we have that $\|h_A\|_2 = \|A\|_2$. Moreover, if $l > 0$, then $\mathbb{E}_{x \sim N(0, \mathbb{I}_d)}[h_A(x)] = 0$.

**Claim 21.** Let $v \in \mathbb{R}^d$ be a unit vector. For $x \in \mathbb{R}^d$, the polynomial $p(x) = \mathrm{He}_l(v \cdot x)$ is harmonic with respect to $x$ with degree $l$. That is, there exists a tensor $A = \mathrm{tensor}(p)$ which is symmetric and with order $l$.

## B.1  Proof of Lemma 15

*Proof.* Recall that we denoted $\lambda^*_{\mathrm{sparse}} = C_1 \cdot \left[ (\ell + C_1 \log \frac{1}{\alpha}) \cdot \log^2(2 + \log \frac{1}{\alpha}) \right]^{2\ell}$ in Algorithm 3.

Observe that if $\left\| \tilde{\Sigma}_U \right\|_F \leq \lambda^*_{\mathrm{sparse}}$, then for any index set $\Omega \subset [d^\ell]$ with $|\Omega| \leq k^\ell$, we have

$$\lambda_{\max}(\tilde{\Sigma}_{\Omega \times \Omega}) \leq \left\| \tilde{\Sigma}_{\Omega \times \Omega} \right\|_F \leq \left\| \tilde{\Sigma}_U \right\|_F \leq \lambda^*_{\mathrm{sparse}},$$

where $\lambda_{\max}(\cdot)$ denotes the maximum eigenvalue and the second step follows from our choice of $U$ which maximizes the restricted Frobenius norm.

Thus, for any $u \in \mathbb{R}^{d^\ell}$ with $\|u\|_0 \leq k^\ell$,

$$u^\top \tilde{\Sigma} u \leq \lambda_{\max}(\tilde{\Sigma}_{\Omega \times \Omega}) \leq \lambda^*_{\mathrm{sparse}}. \tag{B.1}$$

Let $v$ be a $k$-sparse unit vector in $\mathbb{R}^d$. That is, $v \in \mathbb{R}^d, \|v\|_0 \leq k, \|v\|_2 = 1$. Consider some symmetric order-$\ell$ tensor $B$ such that $\mathrm{He}_\ell(v \cdot (x - \mu_T)) = h_B(x - \mu_T)$ (Claim 21). Due to the sparsity of $v$, we know that $B$ is an outer product of $\ell$ number of $k$-sparse vectors; hence $\|B\|_0 \leq k^\ell$. As $h_B(x - \mu_T)$ is a degree-$\ell$ harmonic polynomial and the vector $P_{d,\ell}(x - \mu_T)$ includes all Hermite polynomials with degree exactly $\ell$, we know that we can write $h_B(x - \mu_T) = u_B \cdot P_{d,\ell}(x - \mu_T)$ for some $u_B \in \mathbb{R}^{d^\ell}, \|u_B\|_0 \leq k^\ell$. Thus, we have that

$$\mathbb{E}[h_B(T - \mu_T)^2] = \mathbb{E}[(u_B \cdot P_{d,\ell}(T - \mu_T))^2] = u_B^\top \tilde{\Sigma} u_B \leq \lambda^*_{\mathrm{sparse}} \|u_B\|_2^2 = \lambda^*_{\mathrm{sparse}} \|B\|_2^2.$$

By Fact 20, observe that $\|B\|_2^2 = \mathbb{E}_{x \sim N(\mu_T, \mathbb{I}_d)}[h_B(x - \mu_T)^2] = \ell!$, and thus we have $\mathbb{E}[\mathrm{He}_\ell(v \cdot (T - \mu_T))^2] = \mathbb{E}[h_B(T - \mu_T)^2] \leq \lambda^*_{\mathrm{sparse}} \ell!$.

As a result, we have for any $k$-sparse unit vector $v \in \mathbb{R}^d$ that

$$\mathbb{E}[\mathrm{He}_\ell(v \cdot (S_G \cap T - \mu_T))^2] = \frac{1}{|S_G \cap T|} \sum_{x \in S_G \cap T} \mathrm{He}_\ell(v \cdot (x - \mu_T))^2$$

$$\leq \frac{1}{\alpha \cdot |T|} \sum_{x \in T} \mathrm{He}_\ell(v \cdot (x - \mu_T))^2$$

$$= \frac{1}{\alpha} \cdot \mathbb{E}[\mathrm{He}_\ell(v \cdot (T - \mu_T))^2] \leq \frac{\lambda^*_{\mathrm{sparse}} \cdot \ell!}{\alpha}, \tag{B.2}$$

where the first inequality follows from the condition that $T$ is $\alpha$-good, which, by Definition 10, implies $|S_G \cap T| / |T| \geq \alpha$.

The remaining analysis borrows the proof strategy from [DKS18b]. In particular, we will need the following lemma.

**Lemma 22** (Lemma 3.34 of [DKS18b]). *For any $v \in \mathbb{R}^d$, the polynomial $\mathrm{He}_l(v \cdot (G - \mu_T))$ has mean $(v \cdot (\mu - \mu_T))^l$ and variance at most $2 \max(l, v \cdot (\mu - \mu_T))^{2(l-1)}$.*

Now to ease the notation, write $\theta := v \cdot (\mu - \mu_T)$. By Cantelli's inequality we have

$$\Pr\left[\mathrm{He}_\ell(v \cdot (G - \mu_T)) \geq \theta^\ell - \sqrt{2}\max(\ell, \theta)^{(\ell-1)}\right]$$

$$\geq 1 - \frac{\mathrm{Var}[\mathrm{He}_\ell(v \cdot (G - \mu_T))]}{\mathrm{Var}[\mathrm{He}_\ell(v \cdot (G - \mu_T))] + \mathrm{Var}[\mathrm{He}_\ell(v \cdot (G - \mu_T))]} \geq 1 - \frac{1}{2} = \frac{1}{2}.$$

Since $S_G$ is representative, by Definition 8

$$\Pr\left[\mathrm{He}_\ell(v \cdot (S_G - \mu_T)) \geq \theta^\ell - \sqrt{2}\max(\ell, \theta)^{(\ell-1)}\right] \geq \frac{1}{2} - \frac{\alpha^3}{100} \geq \frac{49}{100}.$$

Since $T$ is $\alpha$-good, due to Definition 10, $|S_G \cap T| / |S_G| \geq 1 - \frac{\alpha}{6} \geq \frac{80}{100}$, we have that

$$\Pr\left[\mathrm{He}_\ell(v \cdot (S_G \cap T - \mu_T)) \geq \theta^\ell - \sqrt{2}\max(\ell, \theta)^{(\ell-1)}\right] \geq \frac{49}{100} - \frac{20}{100} \geq \frac{1}{4}.$$

On the other hand, due to Eq. (B.2), applying Markov's inequality gives that for any $k$-sparse unit vector $v$,

$$\Pr\left[\mathrm{He}_\ell(v \cdot (S_G \cap T - \mu_T)) \geq \sqrt{\frac{4\lambda^*_{\mathrm{sparse}} \cdot \ell!}{\alpha}}\right] \leq \frac{\mathbb{E}[\mathrm{He}_\ell(v \cdot (S_G \cap T - \mu_T))^2]}{\left(\sqrt{\frac{4\lambda^*_{\mathrm{sparse}} \cdot \ell!}{\alpha}}\right)^2}$$

$$\leq \frac{\lambda^*_{\mathrm{sparse}} \cdot \ell!/\alpha}{4\lambda^*_{\mathrm{sparse}} \cdot \ell!/\alpha} = \frac{1}{4}. \tag{B.3}$$

Recall that $\theta = v \cdot (\mu - \mu_T)$. From Eq. (B.1) and (B.3), we have that for any $k$-sparse unit vector $v \in \mathbb{R}^d$,

$$(v \cdot (\mu - \mu_T))^\ell - \sqrt{2}\max(\ell, v \cdot (\mu - \mu_T))^{(\ell-1)} \leq \sqrt{\frac{4\lambda^*_{\mathrm{sparse}} \cdot \ell!}{\alpha}}.$$

Note that $\theta^\ell \leq \sqrt{2}\max(\ell, \theta)^{(\ell-1)}$ only when $\theta \leq 2\ell$, and so we have that for any $k$-sparse unit vector $v \in \mathbb{R}^d$,

$$v \cdot (\mu - \mu_T) \leq 2\ell + \left(\frac{4\lambda^*_{\mathrm{sparse}} \cdot \ell!}{\alpha}\right)^{\frac{1}{2\ell}}$$

$$= O\left(2\ell + \left(\frac{4(C_1 \cdot \left[(\ell + C_1 \log \frac{1}{\alpha}) \cdot \log^2(2 + \log \frac{1}{\alpha})\right]^{2\ell}) \cdot \ell!}{\alpha}\right)^{\frac{1}{2\ell}}\right)$$

$$= O\left(\alpha^{-\frac{1}{2\ell}} \cdot \sqrt{\ell}\left(\ell + \log \frac{1}{\alpha}\right) \cdot \log^2\left(2 + \log \frac{1}{\alpha}\right)\right).$$

By choosing $v = \mathrm{trim}_k(\mu - \mu_T)$ and combining the above with Lemma 39, we complete the proof. $\qquad\square$

## B.2 Analysis of BASICMF

Recall the notations in BASICMF (Algorithm 4): $R = (C_1 \cdot \log(\frac{1}{\alpha}))^{l/2}$, $\gamma = C_0 \cdot \sqrt{\ell \cdot \log(\frac{\ell d}{\alpha \tau})}$, and the length of the interval $[a, b]$, i.e. $b - a$, equals $C_1 \cdot R \cdot \log(2 + \log \frac{1}{\alpha})$. We will need a series of results to prove Theorem 18. First, we note that if BASICMF returns at Step 2, then $T$ must not be $\alpha$-good in view of Lemma 11. Thus we only need to consider the remaining steps. In particular, we divide the output of BASICMF into three cases:

- CASE 1: it returns TBD at Step 5.
- CASE 2: it returns one subset $\{(T', \alpha')\}$ at Step 9.
- CASE 3: it returns two subsets $\{(T_1, \alpha_1), (T_2, \alpha_2)\}$ at Step 14.

We analyze the performance for each case in the following.

### B.2.1 Analysis of CASE 1

**Proposition 23.** *Consider Algorithm 4. If it returns* TBD *and if $T$ is an $\alpha$-good set, then*
$$|\mathbb{E}[p(G)] - \mathbb{E}[p(T)]| \leq O\big((\ell + \log \tfrac{1}{\alpha})^{\frac{l}{2}} \cdot \log(2 + \log \tfrac{1}{\alpha})\big).$$

*Proof.* We first argue that most of the good samples in $T$ have $p(x)$ value close to $\mathbb{E}[p(G)]$.

**Claim 24.** *If $T$ is $\alpha$-good, then the samples $x \in T \cap S_G$ that satisfy $|p(x) - \mathbb{E}[p(G)]| < R$ constitute at least an $\left(\alpha - \frac{\alpha^3}{100}\right)$-fraction of $T$ and an $(1 - \frac{\alpha}{6} - \frac{\alpha^3}{100})$-fraction of $S_G$.*

Next, we claim that if there exists an appropriate interval $[a, b]$ in Step 3, then the mean of $p(G)$ is in the interval $[a - R, b + R]$.

**Claim 25.** *If $T$ is $\alpha$-good, and the interval $[a, b]$ contains at least $(1 - \frac{\alpha}{2})$-fraction of values of $p(x)$ for $x \in T$, then $\mathbb{E}[p(G)] \in [a - R, b + R]$.*

Now by construction, if Algorithm 4 returns TBD, then

$$\mathrm{Var}[p(T)] \leq C_1 \cdot \left(\ell + C_1 \log \frac{1}{\alpha}\right)^l \cdot \log^2 \left(2 + \log \frac{1}{\alpha}\right). \tag{B.4}$$

On the other hand, the interval $[a, b]$ contains at least $(1 - \frac{\alpha}{2})$ fraction of values of $p(x)$ for $x \in T$. Therefore, the contribution of the samples in $[a, b]$ to the variance gives

$$\mathrm{Var}[p(T)] \geq \left(1 - \frac{\alpha}{2}\right) \cdot \max\left\{0, \left|\mathbb{E}[p(T)] - \frac{a+b}{2}\right| - \frac{b-a}{2}\right\}^2. \tag{B.5}$$

To see this, note that $\frac{a+b}{2}$ is the midpoint and $\frac{b-a}{2}$ is the length of interval $[a, b]$. When $\mathbb{E}[p(T)]$ is inside the interval, $\left|\mathbb{E}[p(T)] - \frac{a+b}{2}\right| - \frac{b-a}{2} < 0$ and the variance is lowered bounded by 0. Otherwise, when $\mathbb{E}[p(T)]$ is outside the interval, the distance from any sample in $[a, b]$ to $\mathbb{E}[p(T)]$ is at least $\left|\mathbb{E}[p(T)] - \frac{a+b}{2}\right| - \frac{b-a}{2} \geq 0$.

Moreover, since $b - a \leq O((\log(1/\alpha))^{l/2} \cdot \log(2 + \log(1/\alpha)))$,

$$|\mathbb{E}[p(T)] - (a+b)/2| \leq \frac{b-a}{2} + \sqrt{\mathrm{Var}[p(T)]} = O((\ell + C\log(1/\alpha))^{l/2} \log(2 + \log(1/\alpha))). \tag{B.6}$$

From the Claim 25, we also have

$$|\mathbb{E}[p(G)] - (a+b)/2| \leq \frac{b-a}{2} + R = O((\ell + C\log(1/\alpha))^{l/2} \log(2 + \log(1/\alpha))). \tag{B.7}$$

By the triangle inequality, we have that $|\mathbb{E}[p(G)] - \mathbb{E}[p(T)]| = O((\ell + C\log(1/\alpha))^{l/2} \log(2 + \log(1/\alpha)))$. $\qquad\square$

*Proof of Claim 24.* Since $T$ is $\alpha$-good, and $\mathrm{Var}[p(G)] \leq 1$. By degree-$l$ Chernoff bound (Lemma 40) and definition of representative set (Definition 8), for $R = (C_1 \cdot \log(1/\alpha))^{l/2}$

$$\begin{aligned}
\Pr[|p(S_G) - \mathbb{E}[p(G)]| \geq R] &\leq e^{-\Omega(R^{2/l})} + \frac{\alpha^3}{100k^{2\ell} \cdot \log^l(\frac{\ell d}{\alpha\tau})} \\
&\leq e^{-C \cdot \log(1/\alpha)} + \frac{\alpha^3}{100k^{2\ell} \cdot \log^l(\frac{\ell d}{\alpha\tau})} \\
&= \alpha^C + \frac{\alpha^3}{100k^{2\ell} \cdot \log^l(\frac{\ell d}{\alpha\tau})} \leq \frac{\alpha^3}{100},
\end{aligned}$$

for large enough constant $C > 0$. $\qquad\square$

*Proof of Claim 25.* From Claim 24, at least an $(\alpha - \alpha^3/100)$-fraction of $T$ is $R$-close to $\mathbb{E}[p(G)]$. Also we know that at most an $\frac{\alpha}{2}$-fraction of $T$ are not in $[a, b]$ by the definition of the interval $[a, b]$. Then, there must be at least

$$\left(\alpha - \frac{\alpha^3}{100}\right) - \frac{\alpha}{2} = \frac{\alpha}{2} - \frac{\alpha^3}{100} = \frac{\alpha}{2}\left(1 - \frac{\alpha^2}{50}\right) > 0$$

fraction of samples in $T$ that are in $[a, b]$ and $R$ close to $\mathbb{E}[p(G)]$. Therefore, $\mathbb{E}[p(G)]$ must be in $[a - R, b + R]$. $\qquad\square$

### B.2.2 Analysis of CASE 2

**Lemma 26.** *Consider Algorithm 4. If it reaches Step 7, there must exist a threshold $t > 2R$ satisfying the inequality thereof.*

*Proof.* We will prove this lemma by contradiction. Assume that Algorithm 4 reaches Step 7, but for all $t > 2R$, we have

$$\Pr[\min\{|p(T) - a|, |p(T) - b|\} \geq t] \leq \frac{32}{\alpha} \exp(-(t - 2R)^{2/l}) + \frac{2\alpha^2}{k^{2\ell} \log^l(\frac{\ell d}{\alpha \tau})}.$$

By change of variables, we have that for any $t > 2R + \frac{b-a}{2}$,

$$\Pr\left[\left|p(T) - \frac{a + b}{2}\right| \geq t\right] \leq \frac{32}{\alpha} e^{-\left(t - 2R - \frac{b-a}{2}\right)^{2/l}} + \frac{2\alpha^2}{k^{2\ell} \log^l(\frac{\ell d}{\alpha \tau})}.$$

Note that this inequality only holds non-trivially when $t \geq t_0$ where $t_0 = 2R + \frac{b-a}{2} + (\log \frac{32}{\alpha})^{l/2}$; namely, if $t < t_0$, the right-hand side is at least 1.

By Lemma 13, we have $\max_{x,y \in T} |p(x) - p(y)| \leq 2k^\ell \cdot \gamma^l$, where $\gamma = C_0 \cdot \sqrt{\ell \cdot \log(\frac{\ell d}{\alpha \tau})}$. Also note that the size of the interval $[a, b]$ equals $C_1 \cdot R \cdot \log(2 + \log \frac{1}{\alpha})$ which is less than $k^\ell \cdot \gamma^l$. Therefore,

$$\max_{x \in T}\left|p(x) - \frac{a + b}{2}\right| \leq 3k^\ell \cdot \gamma^l. \tag{B.8}$$

Then, we have that

$$
\begin{aligned}
\mathrm{Var}[p(T)] &\leq \mathbb{E}\left[\left(p(T) - \frac{a + b}{2}\right)^2\right] \\
&= \int_0^\infty \Pr\left[\left(p(T) - \frac{a + b}{2}\right)^2 \geq t^2\right] dt^2 \\
&\stackrel{\varsigma_1}{=} 2\int_0^{3k^\ell \cdot \gamma^l} \Pr\left[\left|p(T) - \frac{a + b}{2}\right| \geq t\right] t\,dt \\
&= 2\int_0^{t_0} \Pr\left[\left|p(T) - \frac{a + b}{2}\right| \geq t\right] t\,dt + 2\int_{t_0}^{3k^\ell \cdot \gamma^l} \Pr\left[\left|p(T) - \frac{a + b}{2}\right| \geq t\right] t\,dt \\
&\leq t_0^2 + 2\int_{t_0}^{3k^\ell \cdot \gamma^l} \left(\frac{32}{\alpha} e^{-\left(t - 2R - \frac{b-a}{2}\right)^{2/l}} + \frac{2\alpha^2}{k^{2\ell} \cdot \log^l(\frac{\ell d}{\alpha \tau})}\right) t\,dt \\
&= t_0^2 + \frac{2\alpha^2}{k^{2\ell} \cdot \log^l(\frac{\ell d}{\alpha \tau})} \cdot 9k^{2\ell} \cdot \gamma^{2l} + \frac{32}{\alpha} \int_{(\log \frac{32}{\alpha})^{l/2}}^\infty e^{-t^{2/l}} \cdot (2t + 4R + b - a)dt \\
&= t_0^2 + 18C_0^2 \cdot \alpha^2 \cdot \ell^l + \frac{32}{\alpha} \int_{\log \frac{32}{\alpha}}^\infty e^{-u} \cdot (2u^{l/2} + 4R + b - a) \cdot \frac{l}{2} \cdot u^{\frac{l}{2} - 1} du \\
&\stackrel{\varsigma_2}{\leq} O\left((\log(1/\alpha))^l \cdot \log^2(2 + \log(1/\alpha))\right) + O\left(2\alpha^2 \cdot \ell^l\right) \\
&\qquad + O((l + \log 1/\alpha)^l \cdot \log(2 + \log 1/\alpha)) \\
&\leq O\left((\ell + \log(1/\alpha))^l \cdot \log^2(2 + \log(1/\alpha))\right),
\end{aligned}
$$

where $\zeta_1$ holds in view of (B.8), and where $\zeta_2$ follows since

$$\frac{32}{\alpha} \int_{\log(\frac{32}{\alpha})}^{\infty} e^{-u} \cdot (2u^{\frac{l}{2}} + 4R + b - a) \cdot \frac{l}{2} \cdot u^{\frac{l}{2}-1} du$$

$$= \frac{32}{\alpha} \int_{\log(\frac{32}{\alpha})}^{\infty} e^{-u} \cdot 2u^{l-1} \cdot \frac{l}{2} du + \frac{32}{\alpha} \int_{\log(\frac{32}{\alpha})}^{\infty} e^{-u}(4R + b - a) \cdot \frac{l}{2} \cdot u^{\frac{l}{2}-1} du$$

$$= \frac{32}{\alpha} \cdot 2 \cdot \frac{l}{2} \int_{\log(\frac{32}{\alpha})}^{\infty} e^{-u} \cdot u^{l-1} du + \frac{32}{\alpha} \cdot (4R + b - a) \cdot \frac{l}{2} \int_{\log(\frac{32}{\alpha})}^{\infty} e^{-u} \cdot u^{\frac{l}{2}-1} du$$

$$\overset{\zeta_3}{\leq} \frac{32}{\alpha} \cdot 2 \cdot \frac{l}{2} \cdot e^{-\log\frac{32}{\alpha}} \cdot \left(\log\frac{32}{\alpha} + l\right)^{l-1} + \frac{32}{\alpha} \cdot (4R + b - a) \cdot \frac{l}{2} \cdot e^{-\log\frac{32}{\alpha}} \cdot \left(\log\frac{32}{\alpha} + \frac{l}{2}\right)^{\frac{l}{2}-1}$$

$$\leq \left(\log\frac{32}{\alpha} + l\right)^l + (4R + b - a) \cdot \left(\log\frac{32}{\alpha} + \frac{l}{2}\right)^{\frac{l}{2}}$$

$$\leq \left(\log\frac{32}{\alpha} + l\right)^l + \left(4\left(C_1 \cdot \log\frac{1}{\alpha}\right)^{l/2} + C_1 \cdot R \cdot \log\left(2 + \log\frac{1}{\alpha}\right)\right) \cdot \left(\log\frac{32}{\alpha} + \frac{l}{2}\right)^{\frac{l}{2}}$$

$$= O\left(\left(l + \log\frac{1}{\alpha}\right)^l \cdot \log\left(2 + \log\frac{1}{\alpha}\right)\right),$$

where $\zeta_3$ is due to the incomplete gamma function (see Claim 3.11 of [DKS18b]), i.e. $\int_x^{\infty} e^{-t} \cdot t^{s-1} dt \leq e^{-x}(x + s)^{s-1}$, for $s \geq 1, x \geq 0$.

In other words, had we not found an appropriate threshold $t > 2R$ at Step 7, Algorithm 4 would have returned at Step 5, which is a contradiction. This completes the proof. $\square$

Once we have verified the existence of such threshold $t$, it is easy to see that the resultant $T'$ is a subset of $T$, and $\alpha' \geq \alpha$ by algebraic calculation. This has been already shown in [DKS18b].

**Lemma 27** (Lemma 3.13 of [DKS18b])**.** *Consider Algorithm 4. If it reaches Step 9, then the output $\{(T', \alpha')\}$ is such that $T' \subset T$ and $\alpha' > \alpha$.*

Next, we show that BASICMF sanitizes the sample set, i.e. it removes more corrupted samples than the uncorrupted ones.

**Lemma 28.** *Consider Algorithm 4. If it reaches Step 9, and if $T$ is $\alpha$-good and $\mathrm{Var}[p(G)] \leq 1$, then the output $\{(T', \alpha')\}$ is such that $T'$ is $\alpha'$-good.*

*Proof.* Due to Algorithm 1, the $\ell_\infty$-distance among all pairs of the samples are bounded. It remains to show $|S_G \cap T'| / |T'| \geq \alpha'$ and $|S_G \cap T'| / |S_G| \geq 1 - \alpha'/6$.

We claim that for any $t > 2R$, the following holds:

$$\Pr[\min\{|p(S_G) - a|, |p(S_G) - b|\} \geq t] \leq 2e^{-(t-R)^{2/l}} + \frac{\alpha^3}{50k^{2\ell} \cdot \log^l(\frac{\ell d}{\alpha\tau})}. \tag{B.9}$$

To see the rationale, we note that by Claim 25, we have $\mathbb{E}[p(G)] \in [a-R, b+R]$. Since $\mathbb{E}[p(G)] - R \leq b$, we have

$$\Pr[p(S_G) - b \geq t] \leq \Pr[p(S_G) - (\mathbb{E}[p(G)] - R) \geq t]$$
$$= \Pr[p(S_G) - \mathbb{E}[p(G)] \geq t - R]$$
$$\leq \Pr[p(G) - E[p(G)] \geq t - R] + \frac{\alpha^3}{100k^{2\ell} \cdot \log^l(\frac{\ell d}{\alpha\tau})}$$
$$\leq e^{-(t-R)^{2/l}} + \frac{\alpha^3}{100k^{2\ell} \cdot \log^l(\frac{\ell d}{\alpha\tau})},$$

where in the third step, we used the fact that $S_G$ is representative (see Definition 8), and in the last step we applied Lemma 40.

The inequality (B.9) follows since $\min\{|p(S_G) - a|, |p(S_G) - b|\} \geq t$ is a subevent of $|p(S_G) - b| > t$.

Since $T$ is $\alpha$-good, we know that a $1 - \frac{\alpha}{6} \geq \frac{1}{2}$ fraction of the samples in $S_G$ is in $S_G \cap T$. Therefore,

$$\Pr[\min\{|p(S_G \cap T) - a|, |p(S_G \cap T) - b|\} \geq t] \leq 4e^{-(t-R)^{2/l}} + \frac{\alpha^3}{25k^{2\ell} \cdot \log^l(\frac{\ell d}{\alpha \tau})}. \quad \text{(B.10)}$$

Due to the inequality of Step 7 in Algorithm 4, we know that the above probability is at least $8/\alpha$ times larger for the samples in $T$. Therefore,

$$
\begin{aligned}
\frac{|S_G \cap T'|}{|T'|} &= \frac{|S_G \cap T'|}{|S_G \cap T|} \frac{|S_G \cap T|}{|T|} \frac{|T|}{|T'|} \\
&\geq \left(1 - \frac{\alpha}{8} \cdot \left(1 - \frac{|T'|}{|T|}\right)\right) \cdot \alpha \cdot \frac{|T|}{|T'|} \\
&\geq \left(\left(1 - \frac{\alpha}{8}\right) \cdot \frac{|T|}{|T'|} + \frac{\alpha}{8}\right) \cdot \alpha \\
&= \alpha',
\end{aligned}
$$

meaning that the remaining fraction of good samples in $T'$ is at least $\alpha'$.

On the other hand, since $|S_G \cap T|/|S_G| \geq 1 - \alpha/6$ and $\left(1 - \frac{\alpha}{8} \cdot \left(1 - \frac{|T'|}{|T|}\right)\right)\alpha = \alpha'|T'|/|T|$, we have

$$
\begin{aligned}
\frac{|S_G \cap T'|}{|S_G|} &= \frac{|S_G \cap T'|}{|S_G \cap T|} \frac{|S_G \cap T|}{|S_G|} \\
&\geq \left(1 - \frac{\alpha}{8} \cdot \left(1 - \frac{|T'|}{|T|}\right)\right) \left(1 - \frac{\alpha}{6}\right) \\
&= \left(1 - \frac{\alpha}{8} \cdot \left(1 - \frac{|T'|}{|T|}\right)\right) - \frac{\alpha'|T'|}{6|T|},
\end{aligned}
$$

thus,

$$
\begin{aligned}
\frac{|S_G \cap T'|}{|S_G|} - \left(1 - \frac{\alpha'}{6}\right) &\geq 1 - \frac{\alpha}{8}\left(1 - \frac{|T'|}{T}\right) - \frac{\alpha'}{6}\frac{|T'|}{|T|} - \left(1 - \frac{\alpha'}{6}\right) \\
&= \left(\frac{\alpha'}{6} - \frac{\alpha}{8}\right)\left(1 - \frac{|T'|}{T}\right) > 0
\end{aligned}
$$

This proves that $T'$ is $\alpha'$-good. $\qquad\square$

We summarize the performance of BASICMF in CASE 2 in the following proposition, which is an immediate combination of Lemma 26, Lemma 27, and Lemma 28.

**Proposition 29.** *Consider Algorithm 4. If it reaches Step 7, there must exist $t > 2R$ that satisfies the inequality of this step, and the algorithm will output $\{(T', \alpha')\}$ with $T' \subset T$ and $\alpha' \geq \alpha$. If, in addition, $T$ is $\alpha$-good and $\mathrm{Var}[p(G)] \leq 1$, then $T'$ is $\alpha'$-good.*

### B.2.3 Analysis of CASE 3

**Lemma 30** (Lemma 3.12 of [DKS18b]). *Consider Algorithm 4. If it reaches Step 12, there must exist a threshold $t$ that satisfy the conditions thereof.*

**Lemma 31** (Lemma 3.14 of [DKS18b]). *Consider Algorithm 4. If it reaches Step 12, then the output $\{(T_1, \alpha_1), (T_2, \alpha_2)\}$ is such that $T_1 \subset T$, $T_2 \subset T$, and $\frac{1}{\alpha_1^2} + \frac{1}{\alpha_2^2} \leq \frac{1}{\alpha^2}$.*

**Lemma 32.** *Consider Algorithm 4. If it reaches Step 12, and if $T$ is $\alpha$-good, then the output $\{(T_1, \alpha_1), (T_2, \alpha_2)\}$ is such that $T_i$ is $\alpha_i$-good for some $i \in \{1, 2\}$.*

*Proof.* Recall that Claim 24 lower bounds the fraction of the good samples (i.e. $x \in S_G \cap T$) that satisfy $|p(x) - \mathbb{E}[p(G)]| < R$. Since $T_1$ and $T_2$ overlap in an interval of length at least $2R$, the good samples must be contained in either one of both two clusters. We will show that the $T_i$ with these good samples (in interval of length $2R$) is $\alpha_i$-good.

Since $T$ is $\alpha$-good, we have $|S_G \cap T| / |T| \geq \alpha$ and $|S_G \cap T| / |S_G| \geq (1 - \alpha/6)$. We want to show that (i) $|S_G \cap T_i| / |T_i| \geq \alpha_i$ and (ii) $|S_G \cap T_i| / |S_G| \geq (1 - \alpha_i/6)$.

To show (i), note that $|S_G \cap T_i| \geq \left(\alpha - \frac{\alpha^3}{100}\right) |T|$ due to Claim 24. Thus,

$$\frac{|S_G \cap T_i|}{|T_i|} = \frac{|S_G \cap T_i|}{|T|} \cdot \frac{|T|}{|T_i|} \geq \left(\alpha - \frac{\alpha^3}{100}\right) \cdot \frac{|T|}{|T_i|} = \alpha_i,$$

where the last transition is by definition.

To show (ii), we only have to show that $\alpha_i/6 \geq \alpha/6 + \alpha^3/100$, i.e. $\alpha_i \geq \alpha + 3\alpha^3/50$. Note that $|T| - |T_i| \geq \frac{\alpha}{4}|T|, \forall i$. Thus, $|T| / |T_i| \geq \frac{1}{1-\alpha/4}$ and we can show that

$$\alpha_i \geq \alpha \cdot \frac{1 - \alpha^2/100}{1 - \alpha/4} \geq \alpha \cdot \frac{100 - \alpha}{100 - 25\alpha} \geq \alpha\left(1 + \frac{24\alpha}{100 - 25\alpha}\right) \geq \alpha\left(1 + \frac{3\alpha^3}{50}\right).$$

This completes the proof. $\qquad\square$

Combining Lemma 30, Lemma 31, and Lemma 32, we immediately have the following.

**Proposition 33.** *Consider Algorithm 4. If it reaches Step 12, then there must exist a threshold $t$ that satisfies the conditions in this step. Moreover, the output $\{(T_1, \alpha_1), (T_2, \alpha_2)\}$ is such that $T_1 \subset T$, $T_2 \subset T$, and $\frac{1}{\alpha_1^2} + \frac{1}{\alpha_2^2} \leq \frac{1}{\alpha^2}$. If, in addition, $T$ is $\alpha$-good, then $T_i$ is $\alpha_i$-good for some $i \in \{1, 2\}$.*

### B.2.4 Proof of Theorem 18

*Proof.* Observe that now Theorem 18 is an immediate result by combining Proposition 23, Proposition 29, and Proposition 33. $\qquad\square$

### B.3 Analysis of HARMONICMF

#### B.3.1 Certifying the varaince of $p$ on $G$

*Proof of Lemma 19.* The proof follows directly from Lemma 3.31 of [DKS18b]. $\qquad\square$

#### B.3.2 Analysis for $p_1$

*Proof of Lemma 16.* First, if at any subroutine of HARMONICMF, it returns "NO" or a list of pairs $\{(T_i, \alpha_i)\}$, ANS $\neq$ TBD. If that is not the case, it means HARMONICMF reaches Step 6 and BASICMF returns "TBD", then we have

$$\mathrm{Var}[p_1(T)/\beta] \leq C_1 \cdot \left(\ell + C_1 \log \frac{1}{\alpha}\right)^\ell \cdot \log^2\left(2 + \log \frac{1}{\alpha}\right),$$

because $p_1$ is of degree $\ell$. However, recall that the condition of Step 6 in Algorithm 3 is satisfied, thus

$$\mathrm{Var}[p_1(T)] = \mathrm{Var}[v^* \cdot P_{d,\ell}(T - \mu_T)] \geq \lambda^* \geq \lambda^*_{\mathrm{sparse}}$$

$$= \left[C_1 \cdot \left(\ell + C_1 \log \frac{1}{\alpha}\right) \cdot \log^2\left(2 + \log \frac{1}{\alpha}\right)\right]^{2\ell}$$

$$\geq \left(C_1 \cdot \left(1 + \log \frac{1}{\alpha}\right) \cdot \log^2\left(2 + \log \frac{1}{\alpha}\right)\right)^\ell \cdot C_1 \cdot \left(\ell + C_1 \log \frac{1}{\alpha}\right)^\ell \cdot \log^2\left(2 + \log \frac{1}{\alpha}\right)$$

$$= \beta^2 \cdot C_1 \cdot \left(\ell + C_1 \log \frac{1}{\alpha}\right)^\ell \cdot \log^2\left(2 + \log \frac{1}{\alpha}\right),$$

which induces a contradition. We conclude that BASICMF will not return "TBD" at Step 6, which completes the proof. $\qquad\square$

#### B.3.3 Analysis for $p_2$

*Proof of Lemma 17.* We see that the lemma holds as long as HARMONICMF returns either "NO" or a list of $(T_i, \alpha_i)$ for $p_2$ correctly. First, we claim that $p_2(x)$ is harmonic such that MULTILINEARMF

multifilters correctly at Step 4. To prove the claim, simply note that $p_2$ is of degree $2\ell$ and consists of a set of $k^{2\ell}$ Hermite polynomials. In addition, $p_2(x)$ only applies on a set of $2\ell k^{2\ell}$ coordinates.

Based on the correctness of MULTILINEARMF, it remains to show that if every subroutine of HARMONICMF returns "TBD", then $T$ must not be $\alpha$-good. Consider that Algorithm 5 reaches Step 6, and BASICMF returns "TBD". Applying Lemma 19, we know that $\mathbb{E}[p_2(G)^2] \leq \beta^2 = \left(C_1 \cdot (1 + \log(\frac{1}{\alpha})) \cdot \log^2(2 + \log(\frac{1}{\alpha}))\right)^{2\ell}$. Therefore, $\mathrm{Var}[\frac{1}{\beta} \cdot p_2(G)] \leq \frac{1}{\beta^2} \cdot \mathbb{E}[p_2(G)^2] \leq 1$ and thus satisfies the preconditions of BASICMF. Then, if BASICMF also returns "TBD", we can show that $\mathrm{Var}[\frac{1}{\beta} \cdot p_2(T)] = O\left((\ell + \log(\frac{1}{\alpha}))^{2\ell} \cdot \log^2(2 + \log(\frac{1}{\alpha}))\right)$ according to Theorem 18. Thus,

$$\mathrm{Var}[p_2(T)] \leq \beta^2 \cdot O(\ell + \log(1/\alpha))^{2\ell} \log^2(2 + \log(1/\alpha))$$
$$\leq O((\ell + \log(1/\alpha)) \log^2(2 + \log(1/\alpha)))^{4\ell}.$$

We then show by contradiction. Assume the above holds and $T$ is $\alpha$-good. Due to Theorem 18, we can show that

$$|\mathbb{E}[p_2(G)] - \mathbb{E}[p_2(T)]| \leq \beta \cdot O(\ell + \log(1/\alpha))^{\ell} \log(2 + \log(1/\alpha))$$
$$\leq O((\ell + \log(1/\alpha)) \log^2(2 + \log(1/\alpha)))^{2\ell}.$$

Additionally, since

$$|\mathbb{E}[p_2(G)]| \leq \sqrt{\mathbb{E}[p_2^2(G)]} \leq \beta^2 = O((\ell + \log(1/\alpha)) \log^2(2 + \log(1/\alpha)))^{2\ell},$$

Therefore, by Cauchy-Schwarz inequality, we conclude that $|\mathbb{E}[p_2(T)]| \leq O((\ell + \log(\frac{1}{\alpha})) \log^2(2 + \log(\frac{1}{\alpha})))^{2\ell}$. However, by construction, we have

$$|\mathbb{E}[p_2(T)]| = \mathbb{E}\left[\mathrm{Tr}\left(\frac{(\tilde{\Sigma})_U}{\left\|(\tilde{\Sigma})_U\right\|_F} \left(P_{d,\ell}(T - \mu_T) P_{d,\ell}(T - \mu_T)^\top\right)\right)\right]$$
$$= \mathrm{Tr}\left((\tilde{\Sigma})_U \tilde{\Sigma}\right) = \left\|(\tilde{\Sigma})_U\right\|_F \overset{\zeta_4}{\geq} \lambda^*_{\mathrm{sparse}}$$
$$\geq C_1 \cdot ((\ell + C_1 \log(1/\alpha)) \log^2(2 + \log(1/\alpha)))^{2\ell},$$

where $\zeta_4$ is due to the condition in Step 4 of Algorithm 3. This is a contradiction.

Hence, we conclude that $T$ cannot be $\alpha$-good and we remove it from the list. Moreover, if BASICMF returns NO or a list $\{(T_i, \alpha_i)\}$, the guarantees follow from Theorem 18. The proof is complete. □

**Lemma 34** (Algorithm 5). *Consider Algorithm 5 with input polynomial being $p_1$ or $p_2$ in view of Algorithm 3, and denote by ANS its output. With probability $1 - \tau$, the following holds. If ANS = NO, then $T$ is not $\alpha$-good. If ANS $= \{(T_i, \alpha_i)\}_{i=1}^m$ for some $m \leq 2$, then $T_i \subset T$ for all $i \in [m]$ and $\sum_{i=1}^m \frac{1}{\alpha_i^2} \leq \frac{1}{\alpha^2}$; if additionally $T$ is $\alpha$-good, then at least one $T_i$ is $\alpha_i$-good.*

*Proof.* Inside any subroutine of BASICMF or MULTILINEARMF called by HARMONICMF, if ANS is assigned "NO" or a list of pairs $\{(T_i, \alpha_i)\}$, the guarantees are ensured by Lemma 19, Theorem 18 and Lemma 36. It remains to show the correctness of the algorithm returning "NO" at Step 6 when BASICMF returns TBD, which is implied by Lemma 17. □

### B.4 Proof of Theorem 14

*Proof.* The theorem follows from Lemma 15, Lemma 16, Lemma 17, Theorem 18 and Lemma 34. □

## C Proof of Theorem 1

Theorem 1 directly follows from the guarantees of our initial clustering step (Lemma 12), the main subroutine (Theorem 14), and the black-box list reduction algorithm (Proposition 37).

*Proof of Theorem 1.* Consider Algorithm 2. By Lemma 12, $T$ will be divided into at most $\frac{1}{2\alpha}$ number of subsets, at least one of which is $\frac{\alpha}{2}$-good. Algorithm 2 then maintains a list $\mathcal{L}$ of pairs $\{(T_i, \alpha_i)\}$ on which Algorithm 3 is called repetitively until the list becomes empty. Theorem 14 implies that when Algorithm 3 is called on some $T_i \in \mathcal{L}$ which is $\alpha_i$-good, if a list of pairs $\{(T_j, \alpha_j)\}$ is returned, then at least one of $\{T_j\}$ is $\alpha_j$-good ($\alpha_j > \alpha_i$). This ensures that there always exists an $\frac{\alpha}{2}$-good subset $T_i$ in list $\mathcal{L}$, except that a leaf node has been created for this branch and the empirical mean of an $\frac{\alpha}{2}$-good data set is returnd. We then argue that Algorithm 2 eventually returns an estimated mean at the branch that includes only $\frac{\alpha}{2}$-good subsets. Since the subsets are $\frac{\alpha}{2}$-good, ANS never equals to NO. In addition, the branch will not create child nodes forever: note that the true multifiltering step is in BASICMF, and both Step 8 and 12 reduce the subset size $|T_i|$ by at least 1; since $\alpha_i$ is non-decreasing, the algorithm cannot remove only inliers; by Definition 10, $|S_G \cap T_i| \geq (1 - \alpha_i/6)\,|S_G| \geq \frac{1}{2}\,|S_G|$. Therefore, the algorithm must return an estimated mean when there is no outliers to filter.

We then bound the list size of the returned list of estimated means. Since during the process of multifiltering, $\sum_i \alpha_i^{-2}$ is non-increasing, we have that $\sum_{i=1}^{|\mathcal{L}|} \alpha_i^{-2} \leq \frac{1}{2\alpha} \cdot \alpha^{-2}$ at any point of Algorithm 2. In addition, $\alpha_i \leq 1, \forall i$, meaning that the list size will never be larger than $O(\alpha^{-3})$. So does the size of $M$. Then, by applying LISTREDUCTION on $M$ with $|M| \leq O(\alpha^{-3})$, the list size can be reduced to $O(\alpha^{-1})$ in view of Proposition 37.

Finally, note that CLUSTER runs in time $O\big(\mathrm{poly}(|T|, d)\big)$, ATTRIBUTE-EFFICIENT-MULTIFILTER runs in time $O\big(\mathrm{poly}(|T|, d^\ell)\big)$ in view of Theorem 14, and LISTREDUCTION runs in time $O\big(\mathrm{poly}(|T|, d)\big)$. Moreover, there are at most $O(|T|/\alpha^3)$ number of calls to ATTRIBUTE-EFFICIENT-MULTIFILTER, and only one call to CLUSTER and one call to LISTREDUCTION, we conclude that the time complexity of Algorithm 2 is $O\big(\mathrm{poly}\big(|T|, d^\ell, \frac{1}{\alpha}\big)\big)$. $\qquad\square$

# D   Omitted Algorithms

In the following, we present the omitted algorithms. In particular, MULTILINEARMF (Algorithms 6) is an important component of HARMONICMF, for which we tailor the algorithms in [DKS18b] to our sparse setting. MULTILINEARMF will further invoke DEGREE2HOMOGENEOUS (Algorithm 7). Algorithm 8, due to [DKS18b], is the black-box list reduction approach that was invoked in Algorithm 2.

## D.1   MULTILINEARMF

We introduce useful facts about multilinear polynomial here. For $d, l \in \mathbb{N}$, a polynomial $p(x_1, \ldots, x_l) : \mathbb{R}^{dl} \to \mathbb{R}$, where $x_i \in \mathbb{R}^d$, is called multilinear if it is linear in each of its $l$ arguments, i.e. if holds that $p(a \cdot x_1 + b \cdot x_1', x_2, \ldots, x_l) = a \cdot p(x_1, x_2, \ldots, x_l) + b \cdot p(x_1, \ldots, x_l)$, for all $a, b \in \mathbb{R}$ and $x_i, x_i' \in \mathbb{R}^d$, and similarly for all the other arguments. Moreover, a polynomial $p$ is called symmetric if $p(x_1, \ldots, x_l) = p(x_{\pi(1)}, \ldots, x_{\pi(l)})$ for any permutation $\pi : [l] \to [l]$. Any degree-$l$ multilinear polynomial $p : \mathbb{R}^{dl} \to \mathbb{R}$ can be expressed as $A(x_1, \ldots, x_l)$ for an order-$l$ tensor $A$ over $\mathbb{R}^d$. Moreover, $A$ is symmetric if $p$ is symmetric.

---

**Algorithm 6** MULTILINEARMF

---

**Require:** A multiset of samples $T \subset \mathbb{R}^d$, parameter $\alpha \in (0, 1/2]$, failure probability $\tau \in (0, 1)$, a degree-$l$ multilinear polynomial $V(x_1, \ldots, x_l)$ over $\mathbb{R}^{dl}$ with $\|V\|_2 \leq 1$, where $V$ is the outer product of $l$ number of $\psi$-sparse vectors.

1: If $l = 1$, run BASICMF on $V(x - \mu_T)$, and **return** its output.

2: Compute the quadratic polynomial $q(x) = \|V(x - \mu_T)\|_2^2$, where $x \in \mathbb{R}^d$ and $Vx$ is an order-$(l - 1)$ tensor with $(Vx)_{i_2, \ldots, i_l} = \sum_{i_1} x_{i_1} V_{i_1, \ldots, i_l}$.

3: Run DEGREE2HOMOGENEOUS on $q(x)$. If it returns NO or a list $\{T_i, \alpha_i\}$, then **return** the same result.

4: Sample a set $\Phi$ of $m = 200 \cdot \alpha^{-1} \log(4/\tau)$ instances uniformly at random from $T$.

5: $\forall x \in \Phi$, let $V_x = \frac{1}{\sqrt{q(x)}} \cdot V(x - \mu_T)$. ANS $\leftarrow$ MULTILINEARMF on $(T, Vx, l - 1, \alpha, \tau/2)$. If it returns NO or a list $\{(T_i, \alpha_i)\}$, then **return** the same result.

6: Otherwise, **return** TBD.

---

**Algorithm 7** DEGREE2HOMOGENEOUS($T, \alpha, \tau, A$)

---

**Require:** A multiset of samples $T \subset \mathbb{R}^d$, parameter $\alpha \in (0, 1/2]$, failure probability $\tau \in (0, 1)$, homogeneous polynomial $x^\top A x$, where $A$ is a $d \times d$ matrix with $\|A\|_* \leq 1$.
1: Compute the $k^2$ largest eigenvalues $\lambda_i$ and eigenvectors $v_i$ of $A$.
2: **for** $i = 1, \ldots, k^2$ **do**
3:     $\text{ANS}_i \leftarrow \text{BASICMF}(T, \alpha, \tau, p)$ with $p(x) = v_i \cdot x$.
4:     **if** $\text{ANS}_i$ is a list of $\{T_i, \alpha_i\}$ or $\text{ANS} = \text{NO}$ **then return** $\text{ANS}_i$.
5: **end for**
6: **if** all $\text{ANS}_i = \text{TBD}$ **then return** TBD.

---

The MULTILINEARMF works in the following way: Given degree-$l$ multilinear polynomial $V(x_1 - \mu_T, \ldots, x_l - \mu_T)$, where $V$ is an order-$l$ symmetric tensor and $x_i$'s are $l$ number of independent variables. The goal is to show that the polynomial has small absolute expectation over $l$ number of i.i.d. draws from $G \sim N(\mu, \mathbb{I}_d)$ if the algorithm does not filter any samples and returns "TBD"; and otherwise, the algorithm multifilters correctly. Since all subroutine of MULTILINEARMF multifilter the data set by calling BASICMF on linear polynomials, we know that if it returns "NO" or a list of pairs $\{(T_i, \alpha_i)\}$, the correctness is guaranteed by Theorem 18. It remains to bound the expected value of the multilinear polynomial when all subroutines return "TBD" and $T$ is $\alpha$-good.

The idea is to sub-sample a large enough sample $\Phi$ from $T$. If $T$ is $\alpha$-good, then with sufficiently high probability, $\exists x \in \Phi$ that is from $G$. By recursively doing this, with sufficiently high probability, we construct a multilinear polynomial $V(G_1 - \mu_T, G_2 - \mu_T, \ldots, G_l - \mu_T)$, the expectation of which is what we concerned about. The upper bound is then shown by induction. When the polynomial is linear, $|\mathbb{E}[V_{x^{l-1}}(G - \mu_T)]| \leq O(\sqrt{1 + \log(1/\alpha)} \cdot \log(2 + \log(1/\alpha))$. Here, we use $V_{x^i}$ to denote taking $i$ times of inner product between tensor $V$ and a vector $x$. Note that $x$ can be different in each time of the inner product. Then, without loss of generality, assume that for order-$(l-1)$ tensor $V_x$, $|\mathbb{E}[V_x(G_1 - \mu_T, \ldots, G_{l-1} - \mu_T)]| \leq f(l-1, \alpha)$, we can show that $|\mathbb{E}[V(G_1 - \mu_T, \ldots, G_l - \mu_T)]| \leq f(l, \alpha)$, provided that $S_G$ is sufficiently epresentative with respect to $G$ on any linear polynomial $V_{x^{l-1}}(x - \mu_T)$ and any quadratic polynomial $q(x) = \|V_{x^i}(x - \mu_T)\|_2^2, \forall i \in [l]$. In this analysis, the only difference between our setting and that of [DKS18b] is the definition of representative set $S_G$ (Definition 8). Fortunately, since all polynomials in our algorithm apply to at most $\psi = 2\ell k^{2\ell}$ coordinates, the linear polynomials must be in $\mathbb{P}(\mathbb{R}^d, 1, 2\ell k^{2\ell}, 2\ell k^{2\ell})$, and the quadratic polynomials must be in $\mathbb{P}(\mathbb{R}^d, 2, 4\ell^2 k^{4\ell}, 2\ell k^{2\ell})$. Therefore, our definition of representative set suffices. The proof follows the same pipeline as that of Lemma 3.27 in [DKS18b]. As a result, it can be shown that $f(l, \alpha) = f(l-1, \alpha) \cdot O(\sqrt{1 + \log(1/\alpha)} \cdot \log(2 + \log(1/\alpha)))$, which renders $|\mathbb{E}[V(G_1 - \mu_T, \ldots, G_l - \mu_T)]| \leq O\big((1 + \log(1/\alpha))^{l/2} \cdot \log^l(2 + \log(1/\alpha))\big)$.

**Definition 35** (Multifilter condition). We say that a list of pairs $\{(T_i, \alpha_i)\}$, where $T_i \subset T$ and $\alpha_i \in (0, 1)$, satisfies the multifilter condition for $(T, \alpha)$ if the following hold:

1. $\sum_i \frac{1}{\alpha_i^2} \leq \frac{1}{\alpha^2}$, and

2. If $T$ is $\alpha$-good, then at least one $T_i$ is $\alpha_i$-good.

**Lemma 36** (MULTILINEARMF, Lemma 3.27 of [DKS18b]). *Given $\alpha \in (0, \frac{1}{2}]$ and $\tau \in (0, 1)$, let $T$ be the input sample set, and a degree-$l$ multilinear polynomial $V(x_1, \ldots, x_l)$ over $\mathbb{R}^{dl}$ with $\|V\|_2 = 1$. Algorithm 6 returns one of the following with guarantees: (1) TBD, and we have that, if $T$ is $\alpha$-good, then with probability $1 - \tau$, $|\mathbb{E}[V(G_1 - \mu_T, \ldots, G_l - \mu_T)]| = O\big((1 + \log(\frac{1}{\alpha}) \log^2(2 + \log(\frac{1}{\alpha})))\big)^{l/2}$, where $G_i$ are independent copies of $G$. (2) NO, then $T$ is not $\alpha$-good. (3) A list of pairs $\{(T_i, \alpha_i)\}$, $T_i \subset T$, satisfying the multifilter condition for $(T, \alpha)$.*

## D.2 LISTREDUCTION

**Proposition 37** (LISTREDUCTION, Proposition B.1 of [DKS18b]). *Fix $\alpha, \beta, \delta, t > 0$ and let $\mu^* \in \mathbb{R}^d$ be finite, and let $S \subseteq T$ be so that (i) $|S| / |T| \geq \alpha$, and (ii) for all unit vectors $v \in \mathbb{R}^d$, we have $\Pr[v \cdot (S - \mu^*) > t] < \delta$. Then, given $M = \{\mu_1, \ldots, \mu_n\} \subset \mathbb{R}^d$ so that $\delta n = o(1)$ and there*

---

**Algorithm 8** LISTREDUCTION$(T, \alpha, \ell, M)$

---

**Require:** A multiset of samples $T \subset \mathbb{R}^d$, parameter $\alpha \in (0, 1/2]$, degree $\ell \geq 1$, a list $M \subset \mathbb{R}^d$.

1: $\beta \leftarrow C_4 \cdot \alpha^{-\frac{1}{2\ell}} \sqrt{\ell}(\ell + \log \frac{1}{\alpha}), \delta \leftarrow \frac{1}{C_5 \log \frac{1}{\alpha}}, t \leftarrow \sqrt{\log(C_5 \log \frac{1}{\alpha})}, n \leftarrow |M|$.

2: For all $\mu_i, \mu_j \in M$, let $v_{ij}$ denote the unit vector in the $\mu_i - \mu_j$ direction.

3: Let $T_i = \cap_{j \neq i} \{x \in T : |v_{ij} \cdot (x - \mu_i)| < \beta + t\}$.

4: $M' \leftarrow \emptyset$.

5: $\forall i \in [n]$, if $|T_i| \geq \alpha(1 - \delta n) |T|$, and $\nexists \mu_j \in M'$ such that $\|\mu_i - \mu_j\|_2 < 2(\beta + t)$, then $M' \leftarrow M' \cup \mu_i$.

6: **return** $M'$.

---

is some $i$ so that $\|\mu_i - \mu^*\|_2 \leq \beta$ for some $\mu_i \in M$, Algorithm 8 outputs $M' \subseteq M$ so that $|M'| \leq \frac{1}{\alpha}(1 + O(\delta n))$ and $\|\mu' - \mu^*\|_2 \leq 3(\beta + t)$ for some $\mu' \in M'$.

**Remark 38.** Under the setting of Algorithm 8, we have that $n = O(\alpha^{-3})$. This combined with the parameter settings in LISTREDUCTION shows that the size of $M'$ is $O(1/\alpha)$ and there is at least one $\mu_i \in M'$ that has comparable error guarantee to those in $M$.

# E   Useful Lemmas

**Lemma 39** (Lemma 3.2 of [CDK$^+$21]). *Fix two vectors $x, y$ with $\|x\|_0 \leq k$ and $\|\mathrm{trim}_k(x - y)\|_2 \leq \delta$. We have that $\|x - \mathrm{trim}_k(y)\|_2 \leq \sqrt{5}\delta$.*

**Lemma 40** (degree-$l$ Chernoff bound, Fact 2.8 of [DKS18b]). *Let $G \sim N(\mu, \mathbb{I}_d)$, $\mu \in \mathbb{R}^d$. Let $p : \mathbb{R}^d \to \mathbb{R}$ be a degree-$l$ polynomial. For any $t > 0$, we have that $\Pr\left[\,|p(G) - \mathbb{E}[p(G)]| \geq t \cdot \sqrt{\mathrm{Var}[p(G)]}\,\right] \leq \exp\left(-\Omega(t^{2/l})\right)$.*

**Lemma 41** (Harmonic and multilinear polynomials, Lemma 3.24 of [DKS18b]). *Let $X, X_{(1)}, \ldots, X_{(\ell)}$ be i.i.d random variables distributed as $N(\mu, \mathbb{I})$ for some $\mu \in \mathbb{R}^d$. Then, for any symmetric matrix $A$, we have*

$$\sqrt{\ell!} \cdot \mathbb{E}[h_A(X)] = \mathrm{Hom}_A(\mu) = \mathbb{E}[A(X_{(1)}, \ldots, X_{(\ell)})],$$

*and*

$$\mathbb{E}[h_A(X)^2] = \sum_{\ell'=0}^{\ell} \left(\binom{\ell}{\ell - \ell'}/\ell'!\right) \cdot \mathrm{Hom}_{B^{(\ell')}}(\mu)$$

*where $B^{(\ell')}$ is the order-$2\ell'$ tensor with*

$$B^{(\ell')}_{i_1,\ldots,i_{\ell'},j_1,\ldots,j_{\ell'}} = \sum_{k_{\ell'+1},\ldots,k_\ell} A_{i_1\ldots i_{\ell'},k_{\ell'+1},\ldots,k_\ell} A_{j_1\ldots j_{\ell'},k_{\ell'+1},\ldots,k_\ell}.$$