# OpenReview forum: "List-Decodable Sparse Mean Estimation"
_NeurIPS.cc/2022/Conference — NeurIPS 2022 Accept_

### Official Review · Reviewer_7URr · 2022-07-06

**Rating:** 7
**Confidence:** 4
**Soundness:** 4 excellent
**Presentation:** 2 fair
**Contribution:** 3 good

**Summary:**


The field of algorithmic robust statistics is concerned with the development of efficient algorithms for statistical estimation problems in settings where the observed data is extremely (often adversarially) noisy. For the canonical problem of mean estimation, a single grossly corrupted data point can completely invalidate the performance of the sample average as a natural estimate of the population mean. In settings usually considered in this domain, one assumes that the algorithm observes $n$ data points generated in the following manner:

1. First $\alpha n$ ``good'' data points are generated from the true underlying distribution $D$.
2. An adversary then inspects the generated samples and adds an arbitrary set of $(1 - \alpha) n$ points to the dataset. Note that the algorithm is given no knowledge of where the corrupted data points are.

Restricting to the specific problem of mean estimation in high dimensions where the data set $X = \left\\{ x_i \right\\}_{i = 1}^n \subset \mathbb{R}^d$, the goal is to recover the mean of the distribution $D$ generating the good data points. This task is made complicated by the fact that the algorithm does not actually know which points these are and natural approaches such as distance based thresholding yield sub-optimal results. In the standard setting when $\alpha \in (1/2, 1]$, approximate identification of $\mu$ is possible with error (in Euclidean norm) ranging between $\sqrt{1 - \alpha}$ to $\alpha$ and computationally and statistically efficient estimators have been designed. In the more challenging list decoding setting when $\alpha < 1/2$, even approximate identification is not possible and one instead returns a list of $1 / \alpha$ estimates one of which is guaranteed to be close to $\mu$. Note that a list of this size is necessary by considering the special case where the data is generated from a mixture of $1 / \alpha$ well behaved distributions. Efficient estimators have also been proposed in this setting with the guarantee that at least one of the elements in the list is close to $\mu$. The degree of closeness ranges between $1 / \sqrt{\alpha}$ when only second moment assumptions are placed on $D$ with improvements possible when stronger restrictions are placed on the distribution. Typically, these involve higher order moments of the distribution and the recovery error correspondingly improves to $(1 / \alpha)^{O (1 / t)}$ if $t$ moments are available.

This paper considers the list decodable setting in the sparse regime where $\mu$ is assumed to be $k$-sparse and $D$ is an isotropic Gaussian centered at $\mu$. In line with prior work on sparse estimation, the goal of the paper is to build an estimator whose sample complexity depends very mildly on the ambient dimension. All prior work incur sample complexity at least $d / \alpha$ which while optimal under no additional assumptions may be too large when the ambient dimension is too large. The paper constructs a polynomial-time estimator which achieves sample complexity $\mathrm{poly} (k, 1 / \alpha, \log d)$ achieving recovery error of $1 / \sqrt{\alpha}$. While the recovery error and sample complexity are sub-optimal (one would expect information theoretic recover error arbitrarily close to $\sqrt{\log (1 / \alpha)}$ and sample complexity $O(k \log (d) / \alpha)$), the number of samples is nearly independent of $d$.

The algorithm is based on the ``multi-filter'' framework for the list-decodable estimators. Intuitively speaking, one starts with a candidate set of good data points and infers one of the following:

- The dataset is well behaved (typically in terms of its moments) and the empirical mean is a good estimate
- Alternative, the dataset is poorly behaved but a certificate of this fact (usually a direction along which moments are not well concentrated) can be used to refine (remove bad points from) the dataset. One then constructs one (when $\alpha > 1/2$) or more (when $\alpha < 1/2$) subsets of the original dataset which are better behaved.

In the multi-filter approach, care must be taken to ensure that we do not create too many datasets or more accurately, the sum of the number of points at any level of this iterative process remains bounded. At the conclusion of this process, the empirical means of all the refined datasets are returned as candidate in the returned list. In the standard list-decoding setting, the certificates belong to the unit vector along the sphere and signify directions along which moments (first and second) deviate significantly from their expected behavior. However, in the sparse setting, it suffices to check only sparse directions -- unit vectors which are $k$-sparse. Since searching for sparse violating directions is typically a hard problem, the paper instead searches for large entries in $\hat{\Sigma} - I$ and uses the $k^2$ largest entries to construct an appropriate certificate.

**** POST REBUTTAL UPDATE ****

The authors have since updated the manuscript with improved results when higher order moments are available at the expense of increased computational and statistical complexity. I've updated my score to reflect these changes.

**Questions:**

Some explanation of the main technical contributions of the paper would be helpful with regards to prior work employing the multi-filter ([DKS 18] for instance). It would be great to have an intuition section detailing the key points of difference from prior work and where sparsity complicates the design of an effective multi-filter.

**Limitations:**

Yes

**Strengths And Weaknesses:**

The design robust algorithms for the setting of sparse estimation is an important problem and the paper makes important progress in this direction conceptually matching the types of improvements that were previously obtained for sparse estimation. My main concern is the lack of technical novelty in the design of the estimator. The algorithmic approach and its subsequent analysis are based on a well-established framework and does not seem significantly novel. For instance, the key lemma (Lemma 12) controlling the behavior of the polynomials used to certify violating directions draw heavily from analogous results in [DKS 18]. Furthermore, the results obtained in the paper are sub-optimal both in terms of recovery guarantees and sample complexity even accounting for the conjectured statistical-computational gap at ($k$ vs $k^2$) for sparse estimation. Despite these drawbacks, the results in the paper are interesting and relevant and would be of interest to the theoretical machine learning and algorithms communities.

---

> ### Author Response · Authors · 2022-08-02
> **Initial Response**
>
> Thank you very much for recognizing the significance of our contribution in obtaining a sparse estimator for the list-decodable statistical problems. We are more than happy to address your concerns with the following responses.
>
> **Q1:** It would be great to have an intuition section detailing the key points of difference from prior work and where sparsity complicates the design of an effective multi-filter.
>
> **Response:** Thank you for pointing out this for us to improve the manuscripts. The challenges introduced by the sparse setting are as follows: 1. Unlike it in the dense setting, the algorithm has access to only poly(k, log d) samples, meaning that the good data is less representative than those in the dense setting (see Def 7). 2. In [DKS18b], the algorithm clusters the points using L_2 distance because the Gaussian samples are naturally concentrated in an L_2 ball of radius sqrt{d}. However, this does not work for the sparse setting: if we were to bound the L_2 distance for every k-sparse support set, the total sample complexity would blow up. This motivated us to appeal to L_infty norm in clustering. 3. The hardness of searching the k-sparse directions complicates the filtering scheme as we are in the dilemma between (a) paying exponential time to iterate through all combinations, and (b) continuing with the current support set (Step 6 of Alg 3) but the algorithm might not filter any samples at all. This is also the necessity of designing Step 7 and proving that it must work given the conditions in Step 3 being violated. We will include these discussions in our future version.
>
>
> **Q2:** The results obtained in the paper are sub-optimal both in terms of recovery guarantees and sample complexity
>
> **Response:** Thank you for the insightful discussion! In our original submission, we aimed to provide the first attribute-efficient algorithm that bears with a sample complexity in poly(k, log d). For the recovery guarantees, a natural extension of our analysis to the degree-\ell polynomial techniques can further improve the estimation error to alpha^{-1/\ell} (for which we have some guarantees included in the revision to be uploaded). As for the sample complexity, we notice that even in the low-dimensional setting, the seminal work of [DKS18] has a sample complexity O(n^4). Given that our algorithm is computationally efficient, the traditional bound O(k^2) was further blowed up due to the search of k^2-sparse directions. We believe that this bound can be further improved if involving new techniques (e.g. a tighter concentration bound), which we are happy to take as our future direction.

---

> > ### Author Response · Authors · 2022-08-08
> > **Any Follow-up Questions?**
> >
> > Hi Reviewer 7URr,
> >
> > We are checking in to see if our initial responses have addressed your concerns and whether you have any follow-up questions. We are delighted to respond to them during the author-reviewer discussion period if any.
> >
> > The manuscripts are updated with the result that improves the error bound to $\alpha^{-1/\ell}$ via degree-$\ell$ polynomials.

---

### Official Review · Reviewer_zY2p · 2022-07-09

**Rating:** 7
**Confidence:** 4
**Soundness:** 3 good
**Presentation:** 3 good
**Contribution:** 2 fair

**Summary:**

The setup for list-decoding of mean estimation (no sparsity yet) is the following:

There is an unknown d-dimensional Gaussian G with mean mu and identity covariance, and an algorithm receives a set of n samples x_1,..., x_n with the promise that alpha * n of the samples are drawn from the unknown Gaussian. The task is to output a list of vectors hat{mu}_1, ..., hat{mu}_l such that one of them is close to mu.

The notable aspects of this setting is that alpha may be significantly smaller than 1/2, so that *most* of the dataset is actually adversarially corrupted. It turns out that the best one can hope for is to output this list of candidate means hat{mu}_1,..., hat{mu}_l, with l = O(1/alpha) such that at least one of them is close to mu. If we want our algorithm to be computationally efficient (i.e., run in time polynomial in the input -- which is sample size * d), then "close" to mu actually depends on alpha.

The first such results along these lines were from [Charikar, Steinhardt, and Valiant (CSV) '17] who got ~O(1/sqrt{alpha}) closeness, and then [Diakonikolas, Kane, Stewart (DKS) '18] improved the closeness parameter to O(1/alpha^{eps}), where the running time of the algorithm is a polynomial of degree O(1/eps). In this setup, the sample complexity is polynomial in d, so the resulting algorithms run in time which is polynomial in d.

This paper considers a very natural scenario, where d is very large, such that polynomial-in-d sample complexity is unacceptable. In this case, one needs an additional assumption on the underlying distribution, and the authors consider the case that mu is k-sparse. This is a very typical assumption to make, and the parametrization in terms of k leads to a sample complexity which is linear in k but logarithmic in d. Hence, the question is whether these list-decodable mean estimation algorithms can be made to run with only poly(k log d) samples.

The main result of this work is an algorithm for achieving this. For any alpha, an algorithm receives poly(k log d) samples, with the promise that an alpha-fraction of the samples come from an identity covariance Gaussian with mean mu which is k-sparse. The algorithm runs in time polynomial in the sample complexity and d, and outputs a list of O(1/alpha) candidate hat{mu}_1,..., hat{mu}_l where one of them is within O(1/sqrt{alpha}) from mu.

As the authors point out, it may be possible to use techniques from [DKS] to improve the error to O(1/alpha^{eps}) with algorithms whose running time is a polynomial in degree O(1/eps).

The approach that this work takes is the 'filtering' approach, which at a high level, proceeds in the following way. First, one comes up with a candidate list of subsets of samples. In each step, an algorithm will argue that either the empirical mean of the subset of samples is good enough for estimation, or else it can remove some samples from the set where the amount of 'false' samples (outliers) removed is much larger than true samples. While the paper proceeds in this approach, there are significant technical challenges in adapting the tools to this setting. Since the number of samples that we have is only poly(klog d), anything which we use samples to estimate will have an error on the order of 1/poly(k log d). The paper consider finding directions to filter outliers by only considering sparse polynomials. Another challenge is that one cannot iterate through all choices of k coordinates in [d], which causes additional technical difficulties.


**Questions:**

I don't think I have any particular questions.

**Limitations:**

The limitations in the theorems are thoroughly discussed.

**Strengths And Weaknesses:**

Strengths:

This is a natural problem and follows a line of work on robust algorithmic statistics. The work is original and clear. It is significant that the techniques developed for 'dense' Gaussians can be adapted for a poly(k log d) dependence when including sparsity.

Weaknesses:

One may consider that this paper leaves the question of 1/alpha^{eps}, involving techniques in [DKS] to future work. In that sense, the paper may be quickly improved.

**** This has since been incorporated! While I cannot verify correctness of the change (I am not an expert in the area), I think that the contribution now contains all elements of a solid paper. The problem is natural, timely, and available avenues have been explored. Therefore, I am changing my score to accept.

---

> ### Author Response · Authors · 2022-08-02
> **Initial Response**
>
> We thank the reviewer for the detailed review of our paper and for endorsing its originality and clarity! Indeed, the main goal of our paper is to design an attribute-efficient algorithm for the list-decodable mean estimation problem in a high-dimensional setting. We didn't focus on the optimality of the error rate. However, we did consider this problem after the submission and derived an improved error bound of O(1/alpha^epsilon) by using degree-1/epsilon polynomials in filtering. We are happy to include this result in our revision and are happy to take any follow-up questions.

---

> > ### Comment · Reviewer_zY2p · 2022-08-05
> > **Thanks for the resonse**
> >
> > Thanks for the response. Given that the authors claimed the techniques can support the improved error bound of O(1/alpha^{eps}), I think this paper should be make a good contribution to the robust statistics literature. Namely, showing that the techniques for list-decodable mean estimation can handle the sparse case in an attribute efficient manner.

---

> > > ### Author Response · Authors · 2022-08-08
> > > **Thank you!**
> > >
> > > We would like to thank the reviewer for appreciating our contribution. The rebuttal revision has been uploaded with the results for higher-degree cases. The reviewer can feel free to check it out at any time.

---

### Official Review · Reviewer_yoWU · 2022-07-11

**Rating:** 5
**Confidence:** 2
**Soundness:** 3 good
**Presentation:** 2 fair
**Contribution:** 3 good

**Summary:**

The authors consider a mean-estimation problem for a sample with (adversarially) corrupted entries.  Assuming that the true mean is sparse, they provide an approximate algorithm, poly-logarithmic in dimension. The proof is based on the use of sparse polynomials.

**Questions:**

It is desirable to explain more about the novelty of the current work. This might includes heuristic ideas about how an efficient algorithm as proposed in the manuscript is available when the entries are corrupted.
Is the key idea to use the Hermite basis?

**Limitations:**

Yes.

**Strengths And Weaknesses:**

Strengths: A concrete, efficient algorithm for the mean estimation is provided. The manuscript is generally well-written.

Weaknesses: The novelty of the algorithm is not clearly explained. It is hard to grasp the improvement since it lacks the comparison with existing algorithms, especially numerical experiments.

---

> ### Author Response · Authors · 2022-08-02
> **Initial Response**
>
> Thank you very much for your valuable comments!
>
> **Q1:** The novelty of the algorithm is not clearly explained.
>
> **Response:** The main contribution of our work is to propose the first polynomial-time algorithm that enjoys sample complexity poly-log in d for the list-decodable mean estimation problem. It is highly non-trivial to design attribute-efficient algorithms in the robust mean estimation regime. Even for the milder corruption setting where alpha>1/2, only until very recently has some guarantees established [BDLS17, DKK+19, CDK+21]. Before this work, all known algorithms for list-decodable learning have sample complexity polynomial in d [CSV17, DKS18b, CMY20, DKK20a, DKK+21a]. The main technical novelty falls in leveraging sparse harmonic polynomials to efficiently filter the outliers. This is because the algorithm is given only poly(k, log d) samples and the estimation thus has an error bound of 1/poly(k, log d) (See Definition 7), which is in stark contrast with all prior works.
>
> In addition, there are several technical challenges when extending the literature to the sparse setting: 1. Searching for the sparse direction where the samples' behavior the most unlike a Gaussian is known as an NP-hard problem. A heuristic algorithm exists when one considers directly applying the algorithm of [DKS18b] to all k-sparse support sets. However, the runtime, sample complexity, and the obtained list size would be exponential, i.e. poly(d^k). 2. When filtering the outliers in list-decodable setting, if the polynomial has degree >1, the traditional techniques fail immediately because the performance of these polynomials relies heavily on how good the estimated mean is (which usually is not good enough as alpha<=1/2). Here, the Hermite basis is the key to filter.
>
> We encourage the reviewer to check out lines 75-108 and also the technical challenges recognized by Reviewer zY2p.
>
> **Q2:** It is hard to grasp the improvement since it lacks the comparison with existing algorithms, especially numerical experiments.
>
> **Response:** The most distinguishing property of our algorithm from the existing ones is that our algorithm works in the high-dimensional setting, where n<<d. As we study the problem from a theoretical perspective and have established provable performance guarantees, we believe experiments are not required.

---

> > ### Comment · Reviewer_yoWU · 2022-08-10
> > **Thank you for the response**
> >
> > Thank you for the response. After reading the response and also other reviews, I would like to adjust my scores. However, I still have doubt on the technical novelty and the presentation for it in the manuscript, which in part also pointed out by Reviewer 7URr. Since I need to evaluate the paper as submitted, I think it is not reasonable to raise the score further.

---

> ### Author Response · Authors · 2022-08-08
> **author reply**
>
> Hi Reviewer yoWU,
>
> Can you look into our initial response and let us know whether we addressed your concern? Since your rating diverges from the other two reviewers (who rated '6' and '7' with confidence '4'), we believe it is better off to communicate with you during the author-reviewer discussion period to clear your questions. Thanks!

---

### Meta-Review · Area_Chair_BkgL · 2022-08-28

**Recommendation:** Accept
**Confidence:** Less certain

**Metareview:**

This paper studies the problem of list-decodable mean estimation under the assumption that the true mean is *sparse* and the clean distribution is Gaussian with identity covariance. In this setting, we are given n data points and a parameter $0<\alpha \leq 1/2$ such that:
(1) an unknown $\alpha$-fraction of the dataset consists of iid samples from $N(\mu, I)$, where the target mean $\mu$ is $k$-sparse (i.e., supported on an unknown set of at most $k$ coordinates), and (2) no assumptions are made on the remaining points. The goal is to output a list of $O(1/\alpha)$ many vectors such that with high probability at least one of these vectors is close to $\mu$, in L2 distance.

This list-decodable mean estimation problem has been well-studied in the dense case (i.e., when $k = d$ where $d$ is the dimension). The authors give an efficient algorithm for the sparse case achieving significantly better sample complexity than in the dense case. The submitted version of the paper achieves error $O(\alpha^{-1/2})$, relying on degree-$2$ polynomials. On August 8, the authors updated their manuscript, achieving improved error using higher degree polynomials. The proposed algorithm (both the initial version and the updated version) uses the multi-filtering technique of Diakonikolas, Kane, Stewart from STOC'18 [DKS18b]. Their approach crucially builds on the multi-filtering technique of [DKS18b] to a degree that the pseudocode of the algorithm and the analysis itself are very similar. On the other hand, the work includes some non-trivial steps to adapt the multi-filtering technique to the sparse setting.

The reviewers eventually agreed that the paper is above the acceptance threshold. The current scores represent the updated scores by the reviewers after the August update of the submission's results. One issue to note here is that the reviewers did not have time to verify (or even read in any detail) the updated version at a technical level; hence, I have low confidence on its correctness.

Overall, the paper seems to be slightly above the acceptance threshold, assuming that the updated version of the paper is correct.

**Award:**

No

---

### Decision · Program_Chairs · 2022-09-14

Accept